# Estimating breakpoints in the Cenozoic Era: An econometric approach

Mikkel Bennedsen<sup>1,2,3</sup>, Eric Hillebrand<sup>1,3</sup>, Siem Jan Koopman<sup>4,5</sup>, Kathrine By Larsen<sup>1,3</sup>, and Rachel Lupien<sup>6</sup>

**Correspondence:** Kathrine By Larsen (kblarsen@econ.au.dk)

Abstract. This study presents a statistical time-domain approach for identifying transitions between climate states, referred to as breakpoints, using well-established econometric tools. Our approach offers the advantage of constructing time-domain confidence intervals for the breakpoints, and it includes procedures to determine how many breakpoints are present in the time series. We apply these tools to a 67.1 million-year-long compilation of benthic foraminiferal oxygen isotopes ( $\delta^{18}$ O), which signify global temperature and ice volume throughout the Cenozoic. This foundational dataset is presented in Westerhold et al. (2020), where the authors use recurrence analysis to identify five breakpoints that define six climate states. Fixing the number of breakpoints to five, our procedure results in breakpoint estimates that closely align with those identified by Westerhold et al. (2020). By allowing the number of breakpoints to vary, we provide statistical justification for more than five breakpoints in the time series. Our method adds to our understanding of Cenozoic climate history in terms of the timing and rate of transitions between climate states and provides a tool for robustly assessing breakpoints in many other paleoclimate time series.

#### 1 Introduction

Understanding the transitions between climate states in Earth's past is crucial for constraining nonlinear and feedback dynamics of our climate system, and anticipating potential climate system responses to anthropogenic warming. The Cenozoic Era, spanning from 66 million years ago (Ma) to today, is particularly informative in this regard, as it is well studied and includes major shifts from hothouse climates with temperatures 10°C warmer than today to the onset of permanent glaciations at both poles (Zachos et al., 2001; Hansen et al., 2013). These transitions, or *breakpoints*, reflect large-scale changes in the climate system, involving shifts in the carbon cycle, ocean circulation, ice volume, and more (Zachos et al., 2008; Mudelsee et al., 2014). As emphasized by Tierney et al. (2020), paleoclimate records are essential for assessing climate sensitivity and evaluating climate models under warmer-than-present conditions. Evidence suggests that the sensitivity of the climate system to external forcings may depend on the climate state (Caballero and Huber, 2013), and that projected future climates may increasingly resemble early Cenozoic conditions under continued emissions (Burke et al., 2018; Steffen et al., 2018). These

<sup>&</sup>lt;sup>1</sup>Department of Economics and Business Economics, Aarhus University, Denmark

<sup>&</sup>lt;sup>2</sup>Aarhus Center for Econometrics, Aarhus University, Denmark

<sup>&</sup>lt;sup>3</sup>Center for Research in Energy: Economics and Markets (CoRE), Aarhus University, Denmark

<sup>&</sup>lt;sup>4</sup>Department of Econometrics and Data Science, Vrije Universiteit Amsterdam, The Netherlands

<sup>&</sup>lt;sup>5</sup>Tinbergen Institute, Amsterdam, The Netherlands

<sup>&</sup>lt;sup>6</sup>Department of Geoscience, Aarhus University, Denmark

insights underscore the importance of identifying when past climate state transitions occurred, how many there were, and how certain we are about their timing. Addressing these questions is crucial for understanding the dynamics of long-term climate variability, and recent work has increasingly emphasized transition detection as a key task in climate data analysis (e.g., Marwan et al., 2021; Trauth, 2025).

A widely used approach to identify breakpoints in paleoclimate records is recurrence analysis, which identifies when a system returns to similar states over time, helping to detect changes in the underlying dynamics of time series (Marwan et al., 2007; Marwan, 2023; see also Fischer et al., 2024; Liang et al., 2025 for recent applications in paleoclimate research). Westerhold et al. (2020) apply this technique to a stacked record of  $\delta^{18}$ O from benthic foraminifera spanning from 67.1 Ma to the present, covering the Cenozoic Era. Based on the recurrence structure of the record, the authors identify four major climate states — Hothouse, Warmhouse, Coolhouse, and Icehouse — which are further divided into six states through time. To conduct this analysis, they resampled the data at an interval of 5 thousand years (kyr) and used both raw and detrended versions. Recurrence analysis provides valuable insights into the recurrence structure and shifts in a time series, and recurrence quantification analysis offers complementary summary measures, such as recurrence rate and determinism. However, the identification of transitions remains largely based on visual interpretation of recurrence plots, and the method lacks formal procedures for determining the number and statistical certainty of the transitions.

Several methodological extensions have sought to address these limitations. For instance, Goswami et al. (2018) propose a breakpoint detection method using a probability density function sequence representation of the time series, which accounts for timestamping uncertainty. Bagniewski et al. (2021) combine recurrence analysis with Kolmogorov–Smirnov tests to statistically assess abrupt shifts in recurrence distributions. Rousseau et al. (2023) apply this method to the Westerhold et al. (2020) data, identifying a similar set of transitions along with several additional ones. As discussed by Marwan et al. (2021), there are several other approaches to identify transitions in paleoclimate time series. Among these, Livina et al. (2010) developed a statistical method of potential analysis and applied it to detect the number of states in an ice core record. In a Bayesian framework, Schütz and Holschneider (2011) develop a method for detecting changes in trend, and Ruggieri (2013) introduce a Bayesian algorithm for identifying multiple breakpoints. Reviews of breakpoint detection techniques in more general climate time series are provided by Reeves et al. (2007) and Lund and Shi (2023).

Recently, Trauth et al. (2024) explored a suite of methods, including recurrence analysis, changepoint detection, and non-linear curve fitting (e.g., sigmoid and ramp functions), to identify climate transitions in a paleoclimate record. They apply a changepoint detection algorithm by Killick et al. (2012), which efficiently detects multiple changepoints in the mean, variance, and trend by minimizing a cost function that balances goodness-of-fit with a penalty for additional changes. The sigmoid functions are characterized by their S-shaped curves and allow for modeling gradual transitions (Crowley and Hyde, 2008; Trauth et al., 2021). In contrast, the ramp functions consist of two horizontal segments connected by a linear trend and represent gradual transitions bounded by abrupt changes in slope, which can be fitted using regression techniques. This method was proposed by Mudelsee (2000) and has been applied to various paleoclimate records (e.g., Fleitmann et al., 2003; Mudelsee and Raymo, 2005). Furthermore, Mudelsee et al. (2014) apply this ramp-function method, among others, to detect major climate transitions in the Cenozoic. For further details, we refer to the textbook treatments in Mudelsee (2014) and Trauth (2025). While these

approaches are widely used, they do not typically include tools for selecting the number of transitions or for assessing the uncertainty in their timing. In this study, we apply an alternative method that addresses both aspects.

Specifically, we employ a statistical approach based on least-squares to estimate breakpoints, and showcase its use with the benthic  $\delta^{18}$ O record from Westerhold et al. (2020). The approach is an econometric time-domain method by Bai and Perron (1998, 2003), which was originally applied to detect shifts in real interest rates data in economics (Garcia and Perron, 1996). We henceforth refer to this as the Bai-Perron framework. While well-established in the econometrics literature, this framework has not been applied to paleoclimate data, where it has a great potential for providing a rigorous statistical foundation for the estimation of breakpoints. In particular, it offers the advantages of constructing confidence intervals for the timestamps of the breakpoints, providing a measure of estimation uncertainty, as well as procedures for selecting the number of breakpoints in the time series. These additional measures are crucial for understanding the certainty, significance, and timing of climate transition periods in the past. The Bai-Perron framework offers flexibility in modeling both abrupt and gradual transitions. We demonstrate its application and benefits by using the data from Westerhold et al. (2020), though the framework is broadly applicable to a wide range of paleoclimate time series.

## 2 Methodology

#### **2.1** Data

We use the dataset provided by Westerhold et al. (2020), which compiles measurements of oxygen isotope ratios from benthic foraminifera across 34 different studies and 14 ocean drilling locations into a single stack covering the Cenozoic. Our study focuses on the benthic  $\delta^{18}$ O record, specifically the correlation-corrected values of benthic  $\delta^{18}$ O.

Benthic  $\delta^{18}$ O measures the deviations in the ratio of the stable oxygen isotopes  $^{18}$ O to  $^{16}$ O in the shells of benthic foraminifera relative to the Vienna Pee Dee Belemnite (VPDB) standard. The ratio of heavy to light stable oxygen isotopes is a function of deep ocean temperatures (Epstein et al., 1951; Shackleton, 1967; Lisiecki and Raymo, 2005) and of the  $\delta^{18}$ O of the seawater in which the foraminifera grow their shells, which in turn is a function of ice volume and salinity (e.g., Waelbroeck et al., 2002; Oerlemans, 2004). Thus, the benthic stack is an important reference record for global climate history across the Cenozoic. Hereafter, we refer to benthic  $\delta^{18}$ O simply as  $\delta^{18}$ O.

The  $\delta^{18}$ O compilation by Westerhold et al. (2020) spans 67.10113 Ma to 564 years before present (Fig. 1). Using recurrence analysis, Westerhold et al. (2020) identify six climate states, and we refer to these as *Warmhouse I* (66-56 Ma), *Hothouse* (56-47 Ma), *Warmhouse II* (47-34 Ma), *Coolhouse I* (34-13.9 Ma), *Coolhouse II* (13.9-3.3 Ma), and *Icehouse* (3.3 Ma-present). Summary statistics for the full record and for each climate state identified by Westerhold et al. (2020) are reported in Appendix B1.

The dataset contains 24,333 entries, of which 74 are missing in the published version. After excluding these, we retain 24,259 data points, ordered from oldest to most recent. The  $\delta^{18}$ O record is irregularly spaced in time, as is typical for paleoclimate

<sup>&</sup>lt;sup>1</sup>These are the values in column "benthic d18O VPDB Corr", found in Sheet 33 of the file aba6853\_tables\_s8\_s34.xlsx provided in the Supplementary Materials of Westerhold et al. (2020).

proxy data. Its average resolution is 2.77 kyr, ranging from 7.28 kyr during Warmhouse II, which has the lowest resolution, to 0.88 kyr during the Icehouse, which has the highest. The longest gap in the data spans about 115.4 kyr and 533 gaps exceed 10 kyr. Additionally, 591 time stamps contain multiple  $\delta^{18}$ O values, with up to four observations recorded at the same time. The  $\delta^{18}$ O stack includes an estimated age uncertainty of  $\pm 100$  kyr in the early Cenozoic and  $\pm 10$  kyr in more recent periods, primarily due to uncertainties in orbital tuning and sedimentation rates (Westerhold et al., 2020). We do not explicitly account for age uncertainty in this study, as it is small relative to the duration of the states we estimate. We therefore expect our main findings to be robust and we will return to this issue in the results section.

**Figure 1.**  $\delta^{18}$ O data from Westerhold et al. (2020). The vertical axis is reversed, following standard practice. The vertical dashed lines show transitions between the climate states by Westerhold et al. (2020). The horizontal axis represents time, measured in millions of years before present. Epoch abbreviations: Cret., Cretaceous; Plio., Pliocene; Pleist., Pleistocene.

#### 95 2.2 The Bai-Perron framework

The Bai-Perron framework is based on minimizing the sum of squared residuals while treating the breakpoints as unknown parameters to be estimated (Bai and Perron, 1998, 2003). Consider a linear regression framework for the dependent variable  $y_t$ , for  $t = 1, \ldots, T$ , and with m breakpoints, corresponding to m + 1 distinct states in the sample. The general model equation is

100 
$$y_t = x_t' \beta + z_t' \delta_j + u_t, \qquad t = T_{j-1} + 1, \dots, T_j,$$
 (1)

with  $j=1,\ldots,m+1$ . The m break dates are denoted by  $(T_1,\ldots,T_m)$ , with the convention that  $T_0=0$  and  $T_{m+1}=T$ , and  $u_t$  is a disturbance term with mean zero and variance  $\sigma_j^2$ . The  $(p\times 1)$ -vector  $x_t$  and the  $(q\times 1)$ -vector  $z_t$  comprise two sets of covariate vectors, for which  $\beta$  is the state-independent vector of coefficients and  $\delta_j$  is the state-dependent vector of coefficients.

Since only specific coefficients are subject to structural breaks, this model is referred to as a partial structural change model.

Moreover, we consider breaks in the variance of  $u_t$  at the break dates  $T_1, \ldots, T_m$ , such that  $\sigma_i^2 \neq \sigma_j^2$  for  $i \neq j$ . The parameters  $\beta$  and  $\delta_i$  are estimated alongside the breakpoints but are not of primary interest here.

We initially treat the number of breakpoints, m, as known and estimate the coefficients and the breakpoints using a sample of T observations of  $\{y_t, x_t, z_t\}$ . The estimation method is based on least squares for both the coefficients and the breakpoints. For each possible set of m breakpoints  $(T_1, \ldots, T_m)$  denoted as  $\{T_i\}_{i=1}^m$ , we obtain estimates of  $\beta$  and  $\delta_j$  by minimizing the sum of squared residuals (SSR), that is,

$$SSR = \sum_{j=1}^{m+1} \sum_{t=T_{j-1}+1}^{T_j} (y_t - x_t'\beta - z_t'\delta_j)^2,$$
(2)

where  $\beta$  is common to all states, while  $\delta_j$  is specific for the state j, which is the period between  $T_{j-1}+1$  and  $T_j$ . The resulting estimated coefficients are denoted as  $\hat{\beta}\left(\left\{T_i\right\}_{i=1}^m\right)$  and  $\hat{\delta}_j\left(\left\{T_i\right\}_{i=1}^m\right)$ . These coefficients are then used to determine the SSR associated with each set of breakpoints,

$$SSR_T(\{T_i\}_{i=1}^m) \equiv \sum_{j=1}^{m+1} \sum_{t=T_{i-1}+1}^{T_j} \left( y_t - x_t' \hat{\beta}(\{T_i\}_{i=1}^m) - z_t' \hat{\delta}_j(\{T_i\}_{i=1}^m) \right)^2.$$
 (3)

The estimated breakpoints are then given by

115 
$$\left(\hat{T}_1, \dots, \hat{T}_m\right) = \underset{T_1, \dots, T_m}{\operatorname{argmin}} SSR_T(\{T_i\}_{i=1}^m).$$
 (4)

The minimization is conducted over all partitions  $(T_1, \ldots, T_m)$  such that  $T_j - T_{j-1} \ge \dim(z_t)$  to ensure that there are enough data points to estimate the parameters  $\delta_j$  in each partition. This procedure leads to estimated parameters for the m breakpoints, i.e.,  $\{\hat{T}_i\}_{i=1}^m$ ,  $\hat{\beta} = \hat{\beta}\left(\{\hat{T}_i\}_{i=1}^m\right)$ , and  $\hat{\delta}_j = \hat{\delta}_j\left(\{\hat{T}_i\}_{i=1}^m\right)$ . Since the possible combinations of the placement of the breakpoints is finite, this optimization can be conducted using a grid search, which can be computationally heavy, especially for many breakpoints. Bai and Perron (2003) introduce an efficient method for determining the global minimizers.

An essential advantage of the Bai-Perron framework is that it allows for constructing confidence intervals for the timing of the breakpoints, something that is not available for the recurrence analysis approach implemented in Westerhold et al. (2020). The construction of confidence intervals is based on the asymptotic distribution of the estimated break dates. The convergence results for the construction of confidence intervals rely on a number of assumptions (see Bai and Perron, 2003).

#### 125 2.3 Model specifications

Three distinct specifications are considered within the Bai-Perron framework, referred to as the "Mean", "Fixed AR", and "AR" models, where AR refers to the autoregressive model of order one with intercept. These are all special cases of the framework outlined in Eq. (1). The simplest among them, the Mean model, is specified as follows,

$$y_t = c_j + u_t, t = T_{j-1} + 1, \dots, T_j,$$
 (5)

for j = 1, ..., m+1, where  $c_j$  is the state-dependent intercept and  $u_t$  is an error term. This model is equivalent to setting  $x_t = 0$ ,  $z_t = 1$ , and  $\delta_j = c_j$  in Eq. (1). A breakpoint in this model specification leads to an abrupt change in the mean of the dependent variable  $y_t$ .

The Fixed AR model extends the Mean model by incorporating an autoregressive term. We obtain the model

$$y_t = c_j + \varphi y_{t-1} + u_t, \qquad t = T_{j-1} + 1, \dots, T_j,$$
 (6)

for  $j=1,\ldots,m+1$ , where  $y_{t-1}$  is the dependent variable lagged by one period, and  $\varphi$  is the autoregressive coefficient that is constant over the whole sample. In this model, the effect of a change in the coefficient  $c_j$  is more gradual, since it depends on the autoregressive dynamics. The Fixed AR model is obtained from Eq. (1) by specifying  $x_t = y_{t-1}$ ,  $\beta = \varphi$ ,  $z_t = 1$ , and  $\delta_j = c_j$ .

The general AR specification also allows the autoregressive term to be state-dependent, resulting in the AR model,

140 
$$y_t = c_j + \varphi_j y_{t-1} + u_t, \qquad t = T_{j-1} + 1, \dots, T_j,$$
 (7)

for  $j=1,\ldots,m+1$ , where the autoregressive coefficient  $\varphi$  in Eq. (6) is now state-dependent and is denoted by  $\varphi_j$ . This model is obtained from Eq. (1) by setting  $x_t=0$ ,  $z_t=(1,y_{t-1})$ , and  $\delta_j=(c_j,\varphi_j)$ . Here, both the intercept and the autoregressive coefficient are state-dependent. Thus, the three specifications are nested: The AR model is the most general, the Fixed AR model is nested in the AR model by setting  $\varphi_1=\varphi_2=\ldots=\varphi_{m+1}=\varphi$ , and the Mean model is nested in the Fixed AR model by setting  $\varphi=0$ .

Figure 2 illustrates how the models capture breakpoints. The Mean model is designed to detect abrupt breaks in the mean of a time series, while the Fixed AR model is for smoother breaks. The AR model is more flexible, allowing for both relatively gradual (e.g.,  $T_1$ ) and abrupt (e.g.,  $T_2$ ) breakpoints compared to the Fixed AR model.

# 2.4 Implementation

145

The Bai-Perron framework is implemented using mbreaks, an R package specifically designed for this purpose (Nguyen et al., 2023). For all model specifications, we set the minimum length of a state, h, to 2.5 million years (Myr), facilitating the estimation of shorter climate states. Also, we let the variance of the error term, denoted as  $\sigma_3^2$ , be state-dependent.

As outlined by Bai and Perron (2003), no serial correlation is permitted in the regression residuals. However, the time series of  $\delta^{18}$ O is likely subject to both autocorrelation and heteroscedasticity, as documented in ice core records (Davidson et al., 2015; Keyes et al., 2023). Autocorrelation occurs when current values correlate with past values, which in paleoclimate data arises from both long-term persistence in climate dynamics (Mudelsee et al., 2014) and taphonomic processes such as bioturbation (Kunz et al., 2020). Since only up to one lag is included in the covariates in the model specifications in this paper, residual serial correlation is likely to remain. Heteroscedasticity, or time-varying error variance, is already partially addressed in the model specifications through state-dependent variance. However, additional heteroscedasticity may arise within the estimated states due to factors such as orbital forcing and changes in ice sheet extent. Addressing both autocorrelation and heteroscedasticity is essential to ensure unbiased parameter estimates and valid confidence intervals for the estimated breakpoints.

Figure 2. Simulated time series using the three model specifications, each with breakpoints  $T_1=25$  and  $T_2=75$ , and total sample size T=100. For the Mean model, we set  $c_1=1.0$ ,  $c_2=1.2$ , and  $c_3=0.8$ . In the Fixed AR model, the parameters are  $\varphi=0.7$ ,  $c_1=0.30$ ,  $c_2=0.36$ , and  $c_3=0.24$ , chosen to yield comparable state-wise means. Likewise in the AR model, we set  $\varphi_1=0.7$ ,  $\varphi_2=0.9$ ,  $\varphi_3=0.4$ ,  $c_1=0.30$ ,  $c_2=0.12$ , and  $c_3=0.48$ . In all specifications, we set  $u_t=0$  for all t.

To account for these issues, we use the autocorrelation and heteroscedasticity consistent (HAC) covariance matrix estimator with prewhitening in the Bai-Perron framework. The prewhitening procedure, proposed by Andrews and Monahan (1992), entails applying an autoregressive model with one lag to  $z_t \hat{u}_t$ , where  $\hat{u}_t$  denotes the residuals. The HAC covariance matrix estimator by Andrews (1991) is then constructed based on the filtered series using the quadratic spectral kernel with bandwidth selected by an AR of order one approximation. This approach is used for all model specifications and is straightforward to implement using the R package (Nguyen et al., 2023).

#### 2.5 Constant data frequency

To conduct breakpoint estimation using the Bai-Perron framework, we need a regularly sampled time series. We use a binning approach to construct a dataset with evenly spaced observations, which is common practice in the analysis of paleoclimate data; see for instance Boettner et al. (2021). We divide the dataset into bins of fixed time intervals and compute the mean of the observations within each bin. In the case of gaps in the binned data, we use the values immediately preceding and succeeding the section with missing data to perform linear interpolation. We consider six different bin sizes, namely 5, 10, 25, 50, 75, and 100 kyr (Fig. 3). Summary statistics for the full sample length and for each climate state identified by Westerhold et al. (2020) for all binning frequencies are provided in Appendix B1.

Data binned at higher frequencies follow the variations in the dataset more closely, whereas data binned at lower frequencies tend to be smoother (Fig. 3). In case of large gaps, a high binning frequency results in linear interpolation between observations (Fig. 3 bottom left). This effect does not occur for periods with many observations, where low binning frequencies capture only a small part of the variation in the original data (Fig. 3 bottom right). Binning offers a simple approach to handle the uneven

frequency of the dataset. However, it leads to data loss at lower binning frequencies and to the introduction of artificial data points resulting from linear interpolation at higher binning frequencies. The selection of binning frequencies can therefore alter the properties of the time series, potentially misrepresenting the dynamics of the original data.

**Figure 3.** Top panel: Benthic foraminiferal  $\delta^{18}$ O data from Westerhold et al. (2020), along with 5 and 100 kyr-binned versions. The record spans 67.10–0.0006 Ma and is based on cores from 14 ocean drilling sites. The vertical dashed lines show transitions between the climate states by Westerhold et al. (2020). Bottom left: 36–35 Ma with an average resolution of approximately 17.2 kyr. Bottom right: 3–2 Ma with an average resolution of approximately 0.9 kyr. Epoch abbreviations: Cret., Cretaceous; Plio., Pliocene; Pleist., Pleistocene.

The Bai-Perron framework is developed for estimating and testing for multiple breakpoints in linear regression models where the regressors are non-trending or state-wise stationary (Bai and Perron, 2003). A time series is considered stationary if its statistical properties, such as mean and variance, do not change over time. The  $\delta^{18}$ O data appears non-stationary over most of the record, even within climate states found by Westerhold et al. (2020). As pointed out by Kejriwal et al. (2013), if the time series maintains its stationarity properties over the respective states, the methods developed for stationary data are still applicable for these cases. However, if the process alternates between stationary and non-stationary states, the theoretical properties of the methodology are unknown.

To investigate whether the time series is non-stationary, we apply the Augmented Dickey-Fuller (ADF) test (Dickey and Fuller, 1979), with the null hypothesis of non-stationarity. For the entire 25 kyr-binned data sample, the ADF test does not reject the null hypothesis at the 1% significance level, indicating non-stationarity. However, when examining the binned data for each climate state identified by Westerhold et al. (2020) separately, the ADF test rejects the null hypothesis at the 1% significance level for the Warmhouse II, Coolhouse I, and Icehouse states. These tests indicate the presence of state-wise nonstationarity, and we therefore need to examine whether the Bai-Perron framework is applicable to data-generating processes that are state-wise non-stationary or alternating between stationary and non-stationary states. For this purpose, we conduct a large simulation study to verify that the Bai-Perron framework works as intended when applied to these types of data-generating processes using the three model specifications. The study is conducted for both independent and identically distributed (i.i.d.) error terms and serially correlated error terms (App. C1 and C2, respectively). The results show that the procedure works well with non-stationarity and is robust to processes with one stationary and one non-stationary state for Fixed AR and AR models. However, the Mean model performs poorly when the data-generating process exhibits high persistence. In the case of serial correlation, the results are less conclusive, but if the states are sufficiently different, the methodology appears effective. The study reveals that the coverage rates for confidence intervals are generally adequate for the Fixed AR model, while the confidence intervals of the AR model are too narrow in many cases. Overall, the Fixed AR model performs best across the data-generating processes considered.

#### 3 Results

# 3.1 Fixed number of breakpoints

As an initial step, we fix the number of breakpoints to 5, which is the number used in the recurrence analysis presented in Westerhold et al. (2020). We estimate the breakpoints and corresponding 95% confidence intervals for each of the binning frequencies, 5, 10, 25, 50, 75, and 100 kyr, using Mean, Fixed AR, and AR models for each (App. B2 and Fig. 4). The estimated confidence intervals around the breakpoints are often asymmetrical. Bai and Perron (2003) advocate the use of asymmetric confidence intervals, as these provide better coverage rates when the data are non-stationary.

For the Mean model, the estimated breakpoints generally remain at the same dates throughout as the binned data frequency decreases step-by-step from 5 kyr to 100 kyr (Fig. 4.a). The width of the 95% confidence intervals increases as the frequency decreases, which can be attributed to the resultant decrease in the number of binned observations available for estimation at the

lower frequencies. All the breakpoints align with those identified by recurrence analysis in Westerhold et al. (2020). A similar pattern of alignment is observed in the Fixed AR model, albeit with tighter confidence intervals (Fig. 4.b). The AR model exhibits more sensitivity to the frequency of the binned data (Fig. 4.c). At higher frequencies, the breakpoints tend to appear in the more recent parts of the sample. However, as the frequency decreases further, the breakpoints are estimated to be in the older parts of the sample period.

For the results using 25 kyr, we find that the estimated breakpoints from the three model specifications align closely with each other and nearly perfectly with those identified by Westerhold et al. (2020). The three model specifications estimated using the 25 kyr-binned data yield parameter estimates that differ across states, reflecting differences in mean and autoregressive dynamics (App. B3).

As a robustness check, we re-estimate the model specifications for 5 breakpoints using the 25 kyr-binned data reversed with respect to the time dimension, so that the time series is ordered from present to past rather than past to present (App. A1). We find that the results of the Mean and Fixed AR models are robust to the ordering of the time axis, with almost unchanged estimated breakpoints. Conversely, the AR model leads to estimated breakpoints in the more recent part of the sample, resulting in breakpoints at 16.9 Ma and 9.7 Ma, which differ from those estimated using the same model and binning frequency with time running forward (Fig. 4).

In summary, changing the binning frequency mainly affects the width of the confidence intervals, while the estimated breakpoint timing remains largely unchanged for both the Mean and Fixed AR models. In contrast, the AR model is more sensitive to resolution and the direction of the time frame. As detailed in the simulation study (App. C), the Mean model fails to accurately detect breakpoints in highly persistent data-generating processes. Consequently, in what follows, we focus on the Fixed AR model for the estimation of breakpoints in the  $\delta^{18}$ O time series. Among the binning frequencies, we proceed with 10 kyr and 25 kyr, as these yield the most consistent results across model specifications and strike a good balance between temporal resolution and signal quality. For the 25 kyr bin width, the mean number of observations per bin is approximately 9, and 3.6 for 10 kyr. However, these numbers vary across the sample, being only 3.5 and 1.4, respectively, in the Warmhouse II and increasing to 28.3 and 11.3, respectively, in the Icehouse period. This highlights the importance of accounting for varying sampling resolution when selecting bin widths. For applications of this framework to other paleoclimate records, we recommend seeking a similar balance.

#### 3.2 Flexible number of breakpoints

We now relax the assumption of a pre-specified number of breakpoints and use information criteria to guide the choice of the number of breakpoints. These criteria are model selection tools that balance goodness of fit with model complexity, helping to avoid overfitting. We initially consider the following three criteria: the Bayesian Information Criterion (BIC) by BIC, the modified Schwarz Information Criterion (LWZ) by Liu et al. (1997), and the modified BIC (KT) by Kurozumi and Tuvaandorj (2011). For all criteria, the preferred number of breakpoints is determined as the number of breakpoints that minimizes the information criterion in question.

Figure 4. A comparison of estimated breakpoints using binned data with frequencies of 5, 10, 25, 50, 75, and 100 kyr from top to bottom, fixing the number of breakpoints to 5 for each model specification. The black dots represent estimated breakpoints, while colored shaded rectangles indicate 95% confidence intervals. The results overlay the  $\delta_{11}^{18}$ O data from Westerhold et al. (2020) (blue dots) and their transitions (vertical dashed lines).

Bai and Perron (2006) note that the BIC and LWZ criteria perform well in the absence of serial correlation, but both lead to overestimation of the number of breakpoints in case of serial correlation in the error term. In simulation studies, we find that the KT information criterion performs poorly, and hence, we exclude it from the subsequent analysis (App. C1 and C2). We also find that the number of breakpoints determined using the Mean model specification is generally too large when employing the information criteria. For the Fixed AR and AR models, the BIC and LWZ criteria typically perform well, especially in datagenerating processes with a large break. With serial correlation in the error term, the BIC criterion tends to slightly overestimate the number of breakpoints, whereas the LWZ criterion generally performs well in the Fixed AR and AR model specifications.

We use the BIC and LWZ information criteria for each model specification and binning frequency to determine the number of breakpoints, and set the minimum state length to h=2.5 Myr (App. B4). For our preferred specification, the Fixed AR model with 25 kyr binning frequency, the LWZ and BIC criteria suggests 6 and 12 breakpoints, respectively. For a 10 kyr binning frequency, the estimated number of breakpoints are 7 and 14, respectively. Thus, the information criteria indicate that the number of distinct climate states in the  $\delta^{18}$ O record is larger than the 5 suggested in Westerhold et al. (2020).

Figure 5. A comparison of estimated breakpoints using the Fixed AR model for one to 15 breakpoints on 25 kyr-binned data. The minimum state length is set to h=1 Myr. The black dots represent estimated breakpoints, while colored shaded rectangles indicate 95% confidence intervals. The results overlay the  $\delta^{18}$ O data from Westerhold et al. (2020) and their transitions.

To further investigate the potential for a higher number of breakpoints, we consider the estimation of up to 15 breakpoints with the minimum length of a state set of h = 1 Myr. This analysis is conducted with the Fixed AR model and 25 kyr-binned data (Fig. 5). These findings show that the breakpoints identified by Westerhold et al. (2020) are preserved in estimations which include 5 or more breakpoints. Furthermore, the additional breakpoints are, in most cases, also very stable and consistently reappear across specifications with a higher number of breakpoints. The same analysis using 10 kyr-binned data led to nearly identical breakpoint estimates, while the Mean and AR models with 25 kyr-binned data yielded estimates that align in certain cases (App. A2, A3, and A4).

Figure 6. A comparison of estimated breakpoints using the Fixed AR model for one and two breakpoints on 5 kyr-binned data for the Icehouse period. The minimum state length is set to h=250 kyr. The black dots represent estimated breakpoints, while colored shaded rectangles indicate 95% confidence intervals. The results overlay the  $\delta^{18}$ O data from Westerhold et al. (2020).

The final estimated breakpoint is placed at 1.425 Ma for the Fixed AR model, just below the upper boundary of the detection window at 1 Ma, imposed by the minimum state length of 1 Myr. Additionally, the estimated breakpoint is located near the midpoint of a linear trend in the time series from approximately 3.3 Ma to the present, suggesting it may be driven by the trend rather than representing a break in the time series (cf. Fig. 5). To investigate this further, we re-estimate the breakpoints for the Fixed AR model, focusing solely on the Icehouse period, with the minimum length of a state set to 250 kyr and 5 kyr binning, leveraging the denser sampling in this part of the record. For the Fixed AR model, the LWZ criterion suggests one breakpoint, while the BIC indicates two. With one breakpoint, the estimate is 1.355 Ma, and with two, the estimated breakpoints are 2.54 Ma and 0.95 Ma (Fig. 6). Estimating more than two breakpoints leads to overlap between the estimated confidence intervals, reducing the interpretability, and these models are therefore excluded. The results are comparable for the Mean and AR models (App. A5 and A6).

#### 3.3 Limitations of the Bai-Perron framework

Although the Bai–Perron framework provides a flexible and well-established method for detecting breaks, it has some limitations. First, the approach assumes piecewise linearity and white noise residuals (Bai and Perron, 2003). However, in the estimations conducted in this study, the residuals are not white noise, indicating that some dynamics are left unexplained. The simulation results show that the Bai-Perron framework nevertheless performs well even when residuals exhibit complex dynamics (App. C). Confidence intervals should still be interpreted with caution. Second, the method is also computationally demanding for high-resolution data, although it remains possible to run on personal computers.

Thirdly, the method does not account for age model uncertainty, which is important for interpreting the timing and significance of time series analytical output (Marwan et al., 2021). In the Westerhold et al. (2020) data, dating uncertainty ranges from about  $\pm 10$  kyr in the younger parts to  $\pm 100$  kyr in the older parts. This can affect the timing of breakpoints and lead

to differences when comparing across records (Franke and Donner, 2019). Previous work has shown that age-depth models often underestimate the true uncertainty in the chronology, which would amplify these effects (Telford et al., 2004). While some progress has been made in including age uncertainty into recurrence analyses (Goswami et al., 2018), incorporating it into the Bai-Perron framework remains a challenge. One could however consider the use of age ensembles which are multiple plausible realizations of the time axis to assess robustness of the estimated breakpoints. Fully integrating age uncertainty into the estimation process, for example by modeling timestamps as random variables, would require further methodological development. However, since the age model uncertainties reported by Westerhold et al. (2020) are small compared to the duration of the estimated climate states, we expect our main findings to be robust.

In addition to age uncertainty, another direction for methodological advancement is developing a breakpoint detection framework for irregularly spaced time series. This would obviate the need for aggregating the data to fixed time intervals, preserving more of the original record. Steps in this direction have already been made in concurrent research (Bennedsen et al., 2024), where the full  $\delta^{18}$ O and  $\delta^{13}$ C stacks (Westerhold et al., 2020) are analyzed while taking the climate state transitions as given and addressing measurement errors.

#### 4 Discussion

Our results demonstrate that the Bai-Perron time-domain framework is a flexible and effective tool for detecting breakpoints in paleoclimate time series. When fixing the number of breakpoints to 5, all model specifications lead to breakpoint estimates that closely match those identified by Westerhold et al. (2020), providing strong statistical support for their climate-state classification. The results of this work are also consistent with the findings by Rousseau et al. (2023).

Information criteria point to a higher number of transitions than previously reported (Westerhold et al., 2020), suggesting the potential for a more detailed classification of Cenozoic climate variability. To explore this, we estimate between 1 and 15 breakpoints using the Fixed AR model (Fig. 5). Using the BIC, we find statistical justification for 12 breakpoints in the time series (Tab. 1). Some of the 12 estimated breakpoints align with major transitions in benthic  $\delta^{13}$ C (Westerhold et al., 2020), atmospheric CO<sub>2</sub> concentration estimates (Hönisch et al., 2023), and global sea level estimates (Miller et al., 2020) (Fig. 7). This alignment may reflect episodes of large-scale reorganization in the Earth system, potentially involving coupled changes in the carbon cycle, temperature, and ice volume.

Five of these breakpoints (BP<sub>3</sub>, BP<sub>5</sub>, BP<sub>7</sub>, BP<sub>9</sub>, and BP<sub>11</sub>) closely match the five major transitions identified by Westerhold et al. (2020), each corresponding to a well-known climatic event. Specifically, BP<sub>3</sub> aligns with the Paleocene–Eocene Thermal Maximum (PETM, 56 Ma), a short-lived but intense global warming event (McInerney and Wing, 2011). BP<sub>5</sub> marks the end of the Early Eocene Climate Optimum (EECO, 47 Ma), a peak in long-term warmth during the early Cenozoic (Westerhold et al., 2018). BP<sub>7</sub> captures the Eocene–Oligocene Transition (EOT, 34 Ma), when Antarctic glaciation began and global temperatures declined sharply (Coxall et al., 2005; Spray et al., 2019). BP<sub>9</sub> corresponds to the Middle Miocene Climate Transition (MMCT, 13.9 Ma), which is associated with expansion of the Antarctic ice sheets (Flower and Kennett, 1994). Finally, BP<sub>11</sub> is close

| Breakpoint name | Estimate | 95% CI           |
|-----------------|----------|------------------|
| $BP_1$          | 61.250   | (61.375, 60.525) |
| $\mathrm{BP}_2$ | 58.200   | (58.275, 57.825) |
| $BP_3$          | 55.975   | (56.275, 55.700) |
| $\mathrm{BP}_4$ | 48.825   | (49.000, 47.675) |
| $\mathrm{BP}_5$ | 46.725   | (46.800, 46.475) |
| $BP_6$          | 39.650   | (39.750, 39.375) |
| $\mathrm{BP}_7$ | 34.025   | (34.050, 33.850) |
| $\mathrm{BP}_8$ | 16.950   | (18.175, 16.225) |
| $\mathbf{BP}_9$ | 13.875   | (13.900, 13.675) |
| $BP_{10}$       | 9.975    | (10.075, 9.700)  |
| $BP_{11}$       | 3.400    | (3.625, 3.325)   |
| $BP_{12}$       | 1.425    | (1.850, 1.225)   |

**Table 1.** Estimated breakpoints and 95% confidence intervals (CI) in Ma for the Fixed AR model with 12 breakpoints determined using the BIC.

to the M2 glaciation event (3.3 Ma), which preceded the onset of sustained Northern Hemisphere glaciation in the Pleistocene (Lisiecki and Raymo, 2005).

Several of the remaining seven estimated breakpoints also coincide with known climate events. For instance, BP<sub>6</sub> aligns with the cooling following the Middle Eocene Climatic Optimum (MECO), originally described by Bohaty and Zachos (2003), and occurs after a peak in atmospheric  $CO_2$  concentrations inferred from boron isotope records (Henehan et al., 2020) (Fig. 8). Another breakpoint, BP<sub>10</sub>, is estimated at 9.975 Ma and broadly coincides with the expansion of  $C_4$  grasslands (Fig. 9), which altered the global carbon cycle and land surface with potential downstream effects on climate (Polissar et al., 2019; Strömberg, 2011). Notably, both of these breakpoints are also identified by Rousseau et al. (2023), who applied recurrence analysis and a Kolmogorov–Smirnov test to the same  $\delta^{18}O$  dataset. BP<sub>8</sub> matches the onset of the Mid-Miocene Climatic Optimum (MMCO), estimated at 16.95 Ma (Flower and Kennett, 1994; Zachos et al., 2001). BP<sub>2</sub> aligns with the maximum in both the  $\delta^{13}C$  and sea level records at 58.03 Ma and 58.21 Ma, respectively (Fig. 7). This period has also been described by Harper et al. (2024) as the peak of the Paleocene Carbon Isotope Maximum (PCIM).

Particularly noteworthy is the lack of breakpoints, even with 15 detections, between the EOT at 34 Ma and the onset of the MMCO around 17 Ma. This is consistent with the idea that this interval is especially stable in the Cenozoic Era, following the establishment of the Antarctic ice sheet (Zachos et al., 2001; Mudelsee et al., 2014).

We now focus on the breakpoints estimated within the Icehouse period, which has an average resolution of 0.88 kyr compared to 2.77 kyr for the full record (Fig. 6). The estimation yields a single breakpoint at 1.355 Ma, which may reflect a midpoint in the record rather than a distinct climatic shift. When allowing for two breakpoints, as suggested by the BIC, they are estimated at 2.54 Ma and 0.95 Ma, coinciding well with the intensification of Northern Hemisphere Glaciation (iNHG)

Figure 7. Overview of key paleoclimate proxies across the Cenozoic Era. From top to bottom: Benthic foraminiferal  $\delta^{18}$ O (Westerhold et al., 2020),  $\delta^{13}$ C (Westerhold et al., 2020), atmospheric CO<sub>2</sub> concentration estimates from multiple proxy records (Hönisch et al., 2023), and global sea-level estimates relative to present (Miller et al., 2020). Breakpoints (black dots) and confidence intervals (light green bars) are estimated using the preferred 12-breakpoint model on the  $\delta^{18}$ O record. The vertical dashed lines show the transitions found by Westerhold et al. (2020). Notable alignments of features in the records with estimated breakpoints include the PETM (56 Ma), EOT (34 Ma), and MMCO (17 Ma), supporting the interpretation of the breakpoints as indicators of major climate transitions.

(Lisiecki and Raymo, 2005) and the Mid-Pleistocene Transition (MPT) (Pisias and Moore, 1981), respectively. The iNHG marks the initiation of sustained, large-scale glaciation in the Northern Hemisphere, beginning around 2.6 Ma, as evidenced by increasing ice-rafted debris (IRD) and declining sea level (McClymont et al., 2023). The MPT marks a change in the periodicity and amplitude of glacial—interglacial cycles, which Clark et al. (2006) describe as a gradual transition occurring between 1.25 and 0.7 Ma. James et al. (2024) provide a dynamical argument supporting this view, while others identify a more abrupt increase in ice volume and deep-ocean cooling centered around 0.9 Ma (Elderfield et al., 2012). Both the iNHG and the MPT are thought to be relatively gradual and complex events, which is supported by the long, asymmetrical confidence intervals, ranging from 2.92 to 2.41 Ma for the first breakpoint and from 1.545 to 0.66 Ma for the second.

These results underscore the capability of the Bai-Perron framework to detect key transitions in Earth's climate history but also emphasize the importance of prior understanding of the climate system when interpreting breakpoint estimates.

**Figure 8.** Boron isotope measurements ( $\delta^{11}$ B) from multiple ocean drilling sites compiled by Henehan et al. (2020), shown alongside  $\delta^{18}$ O values. The estimated breakpoint, BP<sub>6</sub> (black dot), and its confidence interval (light green bar), align with the post-MECO cooling.

**Figure 9.**  $C_4$  grassland expansion, inferred from plant wax carbon isotopes in marine sediments (Polissar et al., 2019), shown alongside  $\delta^{18}O$  values. The estimated breakpoint,  $BP_{10}$  (black dot), and its confidence interval (light green bar), align with this ecological transition.

#### 5 Conclusion

This study presents a statistical time-domain approach to estimate breakpoints in the Cenozoic Era using the econometric tools developed by Bai and Perron (1998, 2003). We analyze the time series of benthic δ<sup>18</sup>O compiled by Westerhold et al. (2020), which is a widely cited foundational record particularly for the field of paleoclimatology. Westerhold et al. (2020) identified 5 breakpoints using recurrence analysis, and our analysis strongly corroborates the placement of these breakpoints across various model specifications and binning frequencies. Our approach offers the advantage of constructing confidence intervals for the ages of the breakpoints, providing a measure of estimation uncertainty. Based on the results of our simulation study, we advocate using the model specification with a state-independent autoregressive term and state-dependent intercept.

By selecting the number of breakpoints using information criteria, we provide statistical justification for more than 5 breakpoints in the time series. For instance, the BIC suggests 12 breakpoints. For these, the 5 transitions identified by Westerhold et al. (2020) are preserved, while the additional breakpoints suggest further divisions of the climate states they found. This points to the potential for a more detailed classification of Cenozoic climate states, adding to our understanding of Earth system dynamics.

Although we focus on the benthic  $\delta^{18}$ O stack (Westerhold et al., 2020) in this study, the Bai–Perron framework is broadly applicable across paleoclimate research and related disciplines. To guide its use in other contexts, we offer several general recommendations based on our findings: 1. Careful consideration should be given to the choice of binning frequency. While finer binning enhances temporal resolution, it may also preserve measurement errors and introduce artifacts by linear interpolations, particularly in unevenly sampled records. Conversely, coarser binning can lead to loss of information. In our application, we

find that the bin width 10 and 25 kyr provide a good balance between detail and signal quality. For other records, we recommend seeking a similar balance. 2. The model specification should reflect the statistical features of the data, such as trends and autocorrelation. Although the Fixed AR model has performed well in our study, the flexibility of the Bai–Perron framework allows users to adapt the model specification to suit different datasets. 3. The number of breakpoints should be selected based on information criteria, such as the BIC or LWZ, which may yield different outcomes depending on model complexity. In our analysis, the BIC tends to favor more breakpoints than the LWZ. We recommend complementing statistical selection with a careful assessment of the climatic relevance of the estimated breakpoints.

These recommendations support the broader application of the framework to other paleoclimate records, like the Cenozoic-spanning reconstructions of  $\delta^{13}$ C or paleo-CO<sub>2</sub>. The method is suitable for detecting both gradual and abrupt transitions, including climatic events such as Dansgaard-Oeschger events (Dansgaard et al., 1993; Livina et al., 2010). In addition to its versatility in application, the framework allows for the inclusion of covariates, opening up many possibilities for future applications. As such, incorporating orbital parameters (e.g., eccentricity, obliquity, and precession; Laskar et al., 2004) could create the potential for detecting transitions while controlling for these external drivers. Additionally, one could investigate breaks in the relationship between orbital forcings and paleoclimate variables, reflecting changes in how strongly these external factors influence climate dynamics. A key example is the MPT, marked by a shift in the dominant glacial cycle from 41 kyr to 100 kyr (Berends et al., 2021; Barker et al., 2025), the timing of which could be estimated using the Bai–Perron framework.

These examples highlight the broader potential of the framework as a flexible tool for paleoclimate data analysis. Understanding when and how breakpoints in the climate system occurred is essential for interpreting past climate variability, events, and shifts, and ultimately for informing projections of future climate change. The Bai–Perron framework provides a statistically rigorous way of estimating these breakpoints, offering new opportunities to deepen our understanding of long-term climate dynamics.

Data availability. The data used in this study are available as the supplementary material of Westerhold et al. (2020).

Code availability. The code used to conduct the analysis is based on the R-package mbreaks by Nguyen et al. (2023) and the implementation is available upon request.

Author contributions. Conceptualization: MB, EH, SJK, KBL. Formal analysis: MB, EH, SJK, KBL, RL. Methodology: MB, EH, SJK, KBL. Software: KBL. Visualization: KBL. Writing (original draft): KBL. Writing (review and editing): MB, EH, SJK, KBL, RL. Authors listed alphabetically.

Acknowledgements. For helpful comments and suggestions, we thank participants at the conferences on Econometric Models of Climate Change (EMCC-VII and VIII) in Amsterdam in 2023 and in Cambridge, UK, in 2024, at the General Assembly of the EGU in Vienna in 2024, and seminar participants at Aarhus University. MB acknowledges funding from the International Research Fund Denmark under grant 7015-00018B.

#### Appendix A: Graphs

#### 400 A1 Reversed time

**Figure A.1.** A comparison of estimated breakpoints using the Mean, Fixed AR, and AR model specifications for five breakpoints on 25 kyr binned data where the time frame is reversed. The black dots represent estimated breakpoints, while colored shaded rectangles indicate 95% confidence intervals. The results overlay the  $\delta^{18}$ O data from Westerhold et al. (2020) and their transitions.

#### A2 One to 15 breakpoints: Fixed AR model 10 kyr

Figure A.2. A comparison of estimated breakpoints using the Fixed AR model for one to 15 breakpoints on 10 kyr binned data. The minimum state length is set to h=1 Myr. The black dots represent estimated breakpoints, while colored shaded rectangles indicate 95% confidence intervals. The results overlay the  $\delta^{18}$ O data from Westerhold et al. (2020) and their transitions.

# A3 One to 15 breakpoints: Mean model

Figure A.3. A comparison of estimated breakpoints using the Mean model for one to 15 breakpoints on 25 kyr binned data. The minimum state length is set to h=1 Myr. The black dots represent estimated breakpoints, while colored shaded rectangles indicate 95% confidence intervals. The results overlay the  $\delta^{18}$ O data from Westerhold et al. (2020) and their transitions.

## A4 One to 15 breakpoints: AR model

Figure A.4. A comparison of estimated breakpoints using the AR model for one to 15 breakpoints on 25 kyr binned data. The minimum state length is set to h=1 Myr. The black dots represent estimated breakpoints, while colored shaded rectangles indicate 95% confidence intervals. The results overlay the  $\delta^{18}$ O data from Westerhold et al. (2020) and their transitions.

# A5 One and two breakpoints in the Icehouse: Mean model 5 kyr

Figure A.5. A comparison of estimated breakpoints using the Mean model for one and two breakpoints on 5 kyr binned data for the Icehouse period. The minimum state length is set to h = 250 kyr. The black dots represent estimated breakpoints, while colored shaded rectangles indicate 95% confidence intervals. The results overlay the  $\delta^{18}$ O data from Westerhold et al. (2020).

# 405 A6 One and two breakpoints in the Icehouse: AR model 5 kyr

Figure A.6. A comparison of estimated breakpoints using the AR model for one and two breakpoints on 5 kyr binned data for the Icehouse period. The minimum state length is set to h=250 kyr. The black dots represent estimated breakpoints, while colored shaded rectangles indicate 95% confidence intervals. The results overlay the  $\delta^{18}$ O data from Westerhold et al. (2020).

# **Appendix B: Tables**

# **B1** Summary statistics: State-wise and full sample

|                 | C                  |        | 0.1   |       |        | D : : :     |
|-----------------|--------------------|--------|-------|-------|--------|-------------|
| Bin size        | State              | Mean   | Sd.   | Max.  | Min.   | Data points |
| 5               | Warmhouse I        | 0.417  | 0.249 | 1.07  | -0.215 | 2221        |
| 5               | Hothouse           | -0.269 | 0.261 | 0.391 | -2.014 | 1800        |
| 5               | Warmhouse II       | 0.897  | 0.366 | 1.894 | -0.254 | 2600        |
| 5               | Coolhouse I        | 2.239  | 0.233 | 2.991 | 1.266  | 4020        |
| 5               | Coolhouse II       | 3.072  | 0.237 | 4.172 | 1.885  | 2120        |
| 5               | Icehouse           | 4.037  | 0.463 | 5.405 | 3.05   | 660         |
| 5               | Full sample period | 1.561  | 1.277 | 5.405 | -2.014 | 13421       |
| 10              | Warmhouse I        | 0.417  | 0.245 | 0.977 | -0.12  | 1111        |
| 10              | Hothouse           | -0.269 | 0.256 | 0.308 | -2.014 | 900         |
| 10              | Warmhouse II       | 0.897  | 0.366 | 1.777 | -0.254 | 1300        |
| 10              | Coolhouse I        | 2.239  | 0.221 | 2.877 | 1.324  | 2010        |
| 10              | Coolhouse II       | 3.072  | 0.228 | 4.122 | 1.975  | 1060        |
| 10              | Icehouse           | 4.034  | 0.447 | 5.33  | 3.181  | 330         |
| 10              | Full sample period | 1.561  | 1.276 | 5.33  | -2.014 | 6711        |
| 25              | Warmhouse I        | 0.418  | 0.237 | 0.912 | -0.065 | 445         |
| 25              | Hothouse           | -0.269 | 0.245 | 0.218 | -1.871 | 360         |
| 25              | Warmhouse II       | 0.898  | 0.358 | 1.688 | 0.01   | 520         |
| 25              | Coolhouse I        | 2.239  | 0.202 | 2.749 | 1.391  | 804         |
| 25              | Coolhouse II       | 3.073  | 0.213 | 3.793 | 2.087  | 424         |
| 25              | Icehouse           | 4.033  | 0.401 | 5.158 | 3.258  | 132         |
| 25              | Full sample period | 1.561  | 1.273 | 5.158 | -1.871 | 2685        |
| 50              | Warmhouse I        | 0.419  | 0.233 | 0.867 | -0.042 | 223         |
| 50              | Hothouse           | -0.268 | 0.233 | 0.197 | -1.871 | 180         |
| 50              | Warmhouse II       | 0.898  | 0.354 | 1.656 | 0.182  | 260         |
| 50              | Coolhouse I        | 2.24   | 0.188 | 2.713 | 1.567  | 402         |
| 50              | Coolhouse II       | 3.072  | 0.206 | 3.72  | 2.156  | 212         |
| 50              | Icehouse           | 4.042  | 0.359 | 4.757 | 3.264  | 66          |
| 50              | Full sample period | 1.562  | 1.271 | 4.757 | -1.871 | 1343        |
| 75              | Warmhouse I        | 0.42   | 0.229 | 0.837 | 0.006  | 148         |
| 75              | Hothouse           | -0.26  | 0.203 | 0.167 | -0.985 | 120         |
| 75              | Warmhouse II       | 0.894  | 0.351 | 1.553 | 0.156  | 173         |
| 75              | Coolhouse I        | 2.239  | 0.181 | 2.717 | 1.691  | 268         |
| 75              | Coolhouse II       | 3.068  | 0.214 | 3.652 | 2.072  | 142         |
| 75              | Icehouse           | 4.041  | 0.351 | 4.753 | 3.283  | 44          |
| 75              | Full sample period | 1.563  | 1.268 | 4.753 | -0.985 | 895         |
| 100             | Warmhouse I        | 0.42   | 0.229 | 0.832 | 0.007  | 112         |
| 100             | Hothouse           | -0.263 | 0.203 | 0.155 | -0.985 | 90          |
| 100             | Warmhouse II       | 0.898  | 0.349 | 1.601 | 0.228  | 130         |
| 100             | Coolhouse I        | 2.241  | 0.175 | 2.685 | 1.739  | 201         |
| 100             | Coolhouse II       | 3.073  | 0.201 | 3.625 | 2.353  | 106         |
| 100             | Icehouse           | 4.047  | 0.344 | 4.673 | 3.4    | 33          |
| 100             | Full sample period | 1.562  | 1.269 | 4.673 | -0.985 | 672         |
| Without binning | Warmhouse I        | 0.428  | 0.25  | 1.07  | -0.215 | 2761        |
| Without binning | Hothouse           | -0.279 | 0.255 | 0.391 | -2.46  | 3030        |
| Without binning | Warmhouse II       | 0.916  | 0.255 | 1.894 | -0.254 | 1786        |
| Without binning | Coolhouse I        | 2.251  | 0.242 | 3.263 | 1.026  | 6669        |
| Without binning | Coolhouse II       | 3.102  | 0.242 | 4.49  | 1.020  | 6282        |
| Without binning | Icehouse           | 4.064  | 0.234 | 5.53  | 2.66   | 3731        |
| Without binning | Full sample period | 2.128  | 1.445 | 5.53  | -2.46  | 24259       |
| C.1. 1: 1.1     |                    | Z.126  |       |       | -2.40  | 24239       |

**Table B.1.** Summary statistics of the binned data with bin sizes (5, 10, 25, 50, 75, and 100 kyr) and the  $\delta^{18}O$  data without binning for each of the states identified by Westerhold et al. (2020) and the full sample period.

# **B2** Estimated breakpoints: 5 breakpoints

| Bin size | BP index |          | Mean             | ]        | Fixed AR         |          | AR              |
|----------|----------|----------|------------------|----------|------------------|----------|-----------------|
|          |          | Estimate | 95% CI           | Estimate | 95% CI           | Estimate | 95% CI          |
| 5        | 1        | 55.965   | (56.085, 55.885) | 55.995   | (56.085, 55.92)  | 33.745   | (33.745, 33.72) |
| 5        | 2        | 46.725   | (46.845, 46.675) | 46.73    | (46.76, 46.68)   | 16.96    | (17.365, 16.78) |
| 5        | 3        | 34.02    | (34.025, 33.915) | 34.05    | (34.075, 34.015) | 13.825   | (13.84, 13.775) |
| 5        | 4        | 13.36    | (13.395, 13.325) | 13.41    | (13.465, 13.34)  | 9.555    | (9.585, 9.505)  |
| 5        | 5        | 2.735    | (2.845, 2.715)   | 2.74     | (3.1, 2.715)     | 3.36     | (3.815, 3.355)  |
| 10       | 1        | 55.97    | (56.15, 55.79)   | 55.99    | (56.15, 55.88)   | 33.77    | (33.77, 33.72)  |
| 10       | 2        | 46.73    | (46.84, 46.64)   | 46.73    | (46.77, 46.64)   | 17.88    | (18.32, 17.64)  |
| 10       | 3        | 34.02    | (34.03, 33.9)    | 34.15    | (34.18, 34.09)   | 13.82    | (13.84, 13.75)  |
| 10       | 4        | 13.36    | (13.4, 13.3)     | 13.82    | (13.89, 13.72)   | 9.59     | (9.72, 9.45)    |
| 10       | 5        | 2.73     | (2.81, 2.7)      | 2.74     | (3.18, 2.71)     | 2.74     | (2.88, 2.72)    |
| 25       | 1        | 55.975   | (56.3, 55.1)     | 56.025   | (56.575, 55.7)   | 55.825   | (55.85, 55.675) |
| 25       | 2        | 46.725   | (47.3, 46.55)    | 46.725   | (46.825, 46.45)  | 48.35    | (48.625, 47.85) |
| 25       | 3        | 34.025   | (34.05, 33.5)    | 34.15    | (34.225, 34.0)   | 33.75    | (33.75, 33.675) |
| 25       | 4        | 13.4     | (13.525, 13.275) | 13.875   | (13.975, 13.65)  | 13.875   | (14.05, 13.55)  |
| 25       | 5        | 2.725    | (2.8, 2.625)     | 2.775    | (3.075, 2.7)     | 2.575    | (2.6, 2.55)     |
| 50       | 1        | 55.95    | (56.2, 54.6)     | 56       | (57.1, 55.35)    | 56       | (56.65, 55.7)   |
| 50       | 2        | 46.7     | (48.15, 46.45)   | 47.1     | (47.25, 46.55)   | 48.8     | (49.1, 40.45)   |
| 50       | 3        | 34.05    | (34.05, 32.8)    | 34.2     | (34.3, 33.9)     | 33.75    | (33.75, 33.6)   |
| 50       | 4        | 13.8     | (14.15, 13.6)    | 13.85    | (14.0, 13.45)    | 16.95    | (17.35, 16.7)   |
| 50       | 5        | 2.75     | (2.9, 2.5)       | 3.15     | (3.4, 3.0)       | 14.3     | (14.55, 12.8)   |
| 75       | 1        | 55.95    | (56.325, 53.775) | 56.25    | (57.45, 54.75)   | 55.95    | (56.325, 55.5)  |
| 75       | 2        | 46.725   | (50.625, 46.425) | 47.1     | (47.475, 46.425) | 53.325   | (53.625, 50.1)  |
| 75       | 3        | 34.05    | (34.05, 30.9)    | 34.2     | (34.425, 33.675) | 34.05    | (34.05, 33.825) |
| 75       | 4        | 13.35    | (13.8, 12.975)   | 13.875   | (14.1, 13.125)   | 16.95    | (17.325, 16.5)  |
| 75       | 5        | 2.775    | (3.375, 2.4)     | 3.15     | (3.525, 2.925)   | 14.475   | (15.075, 14.25) |
| 100      | 1        | 56       | (56.4, 54.0)     | 56.2     | (57.7, 54.5)     | 56       | (56.3, 55.5)    |
| 100      | 2        | 46.7     | (52.5, 46.3)     | 47.1     | (47.7, 46.3)     | 53.4     | (53.8, 52.1)    |
| 100      | 3        | 34.1     | (34.1, 29.4)     | 34.2     | (34.5, 33.4)     | 49.1     | (50.8, 48.8)    |
| 100      | 4        | 13.8     | (14.7, 13.4)     | 13.9     | (14.1, 12.9)     | 34.1     | (34.1, 33.8)    |
| 100      | 5        | 2.9      | (4.2, 2.3)       | 3.4      | (3.8, 3.2)       | 13.8     | (15.7, 12.9)    |

**Table B.2.** Estimated breakpoints and their 95% confidence intervals (in Ma) where the number of breakpoints is fixed to 5, and all values are rounded to three decimals. The table shows estimates for each method across bin sizes 5, 10, 25, 50, 75, and 100 kyr.

# B3 Estimated parameters: 5 breakpoints and 25 kyr binned data

|                                                                | Mean     |       | Fixe     | d AR  |          | AR    |
|----------------------------------------------------------------|----------|-------|----------|-------|----------|-------|
| Parameter                                                      | Estimate | SE    | Estimate | SE    | Estimate | SE    |
| $c_1$                                                          | 0.418    | 0.051 | 0.069    | 0.008 | -0.001   | 0.026 |
| $c_2$                                                          | -0.256   | 0.040 | -0.043   | 0.007 | -0.108   | 0.015 |
| $c_3$                                                          | 0.911    | 0.072 | 0.153    | 0.013 | 0.028    | 0.007 |
| $c_4$                                                          | 2.247    | 0.017 | 0.373    | 0.031 | 0.660    | 0.061 |
| $c_5$                                                          | 3.119    | 0.027 | 0.519    | 0.043 | 0.421    | 0.073 |
| $c_6$                                                          | 4.140    | 0.051 | 0.698    | 0.057 | 2.423    | 0.326 |
| $\varphi$                                                      | ×        | ×     | 0.833    | 0.014 | ×        | ×     |
| $arphi_1$                                                      | ×        | ×     | ×        | ×     | 0.990    | 0.054 |
| $arphi_2$                                                      | ×        | ×     | ×        | ×     | 0.631    | 0.037 |
| $\varphi_3$                                                    | ×        | ×     | ×        | ×     | 0.970    | 0.008 |
| $\varphi_4$                                                    | ×        | ×     | ×        | ×     | 0.706    | 0.027 |
| $arphi_5$                                                      | ×        | ×     | ×        | ×     | 0.865    | 0.024 |
| $arphi_6$                                                      | ×        | ×     | ×        | ×     | 0.419    | 0.081 |
| $\sigma_1^2$                                                   | 0.237    | ×     | 0.095    | ×     | 0.106    | ×     |
| $\sigma_2^2$                                                   | 0.255    | ×     | 0.154    | ×     | 0.140    | ×     |
| $ \sigma_2^2  \sigma_3^2  \sigma_4^2  \sigma_5^2  \sigma_6^2 $ | 0.347    | ×     | 0.112    | ×     | 0.107    | ×     |
| $\sigma_4^2$                                                   | 0.210    | ×     | 0.141    | ×     | 0.140    | ×     |
| $\sigma_5^2$                                                   | 0.208    | ×     | 0.111    | ×     | 0.116    | ×     |
| $\sigma_6^2$                                                   | 0.351    | ×     | 0.340    | ×     | 0.315    | ×     |

**Table B.3.** Estimated parameters and their corresponding standard errors (SE) for each model specification. Parameters absent in a given model specification are denoted by  $\times$ . The number of breakpoints is set to 5, and the parameters are estimated with a binning frequency of 25 kyr and h = 2.5 Myr. All values are rounded to three decimals.

## 410 B4 The number of breakpoints selected by information criteria

| Bin size | N   | Iean | Fix | ked AR | A   | R   |
|----------|-----|------|-----|--------|-----|-----|
|          | BIC | LWZ  | BIC | LWZ    | BIC | LWZ |
| 5        | 19  | 17   | 17  | 7      | 15  | 5   |
| 10       | 17  | 17   | 14  | 7      | 14  | 3   |
| 25       | 17  | 14   | 12  | 6      | 8   | 3   |
| 50       | 17  | 14   | 10  | 0      | 7   | 0   |
| 75       | 17  | 14   | 6   | 0      | 5   | 0   |
| 100      | 17  | 12   | 6   | 0      | 5   | 0   |

**Table B.4.** The number of breakpoints selected using BIC and LWZ criterion for all models and binning frequencies considered. The minimum state length is set to h = 2.5 Myr and the maximum number of breakpoints is 26.

#### **Appendix C: Simulation study**

435

#### C1 Serially uncorrelated error term

In this appendix, we assess whether the methodology by Bai and Perron (1998, 2003) can be used to accurately estimate the number and timing of breakpoints in a state-wise non-stationary time series. We conduct 1000 simulations for each data-generating process (DGP) with a sample size of 500. All the DGPs considered have the following form,

$$y_{t} = c_{1} + \varphi_{1} y_{t-1} + \varepsilon_{t}, \quad \varepsilon_{t} \stackrel{i.i.d.}{\sim} \mathcal{N}\left(0, \sigma^{2}\right) \quad \text{for } t \leq T/2$$

$$y_{t} = c_{2} + \varphi_{2} y_{t-1} + \varepsilon_{t}, \quad \varepsilon_{t} \stackrel{i.i.d.}{\sim} \mathcal{N}\left(0, \sigma^{2}\right) \quad \text{for } t > T/2. \tag{C1}$$

Hence, we consider a single breakpoint in the middle of the sample interval, namely at t = 250. We examine eight DGPs, each specified and described in Table C.1.

| DGP | $\sigma$ | $c_1$ | $c_2$ | $\varphi_1$ | $\varphi_2$ | Description                                                  |
|-----|----------|-------|-------|-------------|-------------|--------------------------------------------------------------|
| 1   | 1        | 0.1   | 0.2   | 1           | 1           | Small break in the drift term of a RW                        |
| 2   | 1        | 0.1   | 1     | 1           | 1           | Large break in the drift term of a RW                        |
| 3   | 1        | 0.1   | 1     | 0.95        | 0.95        | Large break in the intercept and a fixed AR-coefficient      |
| 4   | 1        | 0.1   | 1     | 0.95        | 1           | Break in the intercept and small break in the AR-coefficient |
| 5   | 1        | 0.1   | 1     | 0.5         | 1           | Break in the intercept and large break in the AR-coefficient |
| 6   | 1        | 1     | 1     | 1           | 1           | RW with a drift without a breakpoint                         |
| 7   | 0.5      | 0.1   | 1     | 1           | 1           | Large break in the drift of a RW with low variance           |
| 8   | 1        | 0.1   | 1     | 0.5         | 0.5         | Large break in the intercept and a low fixed AR-coefficient  |

Table C.1. Data-generating processes for the simulation study and short descriptions. RW: random walk.

The DGPs range from random walk models with a break in the drift term to models with breaks in both the intercept and the AR coefficient. For comparison, we include a random walk without breakpoints as the sixth model. For each of the DGPs, we are interested in the performance of the methodology by Bai and Perron (1998, 2003) in estimating the breakpoint and confidence intervals. The model specifications from Section 2.3 are estimated on the data generated by the DGPs, and we use the implementation outlined in Section 2.4. We use the R-package *mbreaks* by Nguyen et al. (2023), and we impose a single breakpoint in the estimation. The left and right panels of Figs. ?? through C.8 display realizations of the DGP and density plots of the estimated breakpoints for each of the models, respectively. The results are summarized in Table C.2, which provides the mean of the estimated breakpoints, and medians of the lower and upper boundaries of the estimated 95% CIs are tabulated along with their coverage rates for each model and DGP.

In the first DGP, a random walk with a small drift term break, we observe that the mean of the estimated breakpoints is later than the true breakpoint in all model specifications. Additionally, the density plots exhibit asymmetry around the true breakpoint. This is expected due to the low magnitude of the break in the drift term, which creates a subtle change in the overall stochastic trend, making accurate breakpoint detection difficult. In the second DGP with a larger drift term break, the estimated breakpoints exhibit a narrower and more bell-shaped density. The mean estimated breakpoints for the Fixed AR and AR models slightly precede the true breakpoint. However, the Mean model performs poorly, with the mean of the estimated breakpoints far from the true breakpoint.

In the third DGP, both the Fixed AR and AR models produce mean estimated breakpoints slightly later than the true breakpoint. The Mean model exhibits better performance in this DGP than in the second DGP. The fourth DGP has a break in the intercept and the AR-coefficient

| DGP |         | Mean  |       |          |         | Fixed AR |       |          |         | AR    |       |          |  |
|-----|---------|-------|-------|----------|---------|----------|-------|----------|---------|-------|-------|----------|--|
|     | BP est. | Lower | Upper | Coverage | BP est. | Lower    | Upper | Coverage | BP est. | Lower | Upper | Coverage |  |
| 1   | 301     | 174   | 655   | 57.1%    | 251     | 216      | 336   | 43.4%    | 290     | 240   | 316   | 22.7%    |  |
| 2   | 333     | -386  | 332   | 95.4%    | 249     | 237      | 262   | 93%      | 249     | 236   | 256   | 77.2%    |  |
| 3   | 263     | 253   | 284   | 41.4%    | 256     | 239      | 260   | 89.9%    | 251     | 241   | 260   | 85.9%    |  |
| 4   | 340     | -190  | 340   | 97.5%    | 249     | 239      | 260   | 95.8%    | 249     | 238   | 250   | 65.8%    |  |
| 5   | 340     | -114  | 340   | 97.1%    | 250     | 239      | 258   | 97%      | 250     | 241   | 250   | 72.9%    |  |
| 6   | 249     | -3325 | 3976  | ×        | 253     | 142      | 371   | ×        | 254     | 202   | 312   | ×        |  |
| 7   | 333     | -282  | 330   | 92%      | 249     | 246      | 253   | 97.8%    | 249     | 246   | 253   | 96%      |  |
| 8   | 249     | 237   | 264   | 95.1%    | 248     | 236      | 263   | 95.2%    | 248     | 236   | 263   | 94.5%    |  |

**Table C.2.** Mean of the estimated breakpoints and medians of the lower and upper boundary of the estimated confidence intervals, along with the coverage rates for each model specification and DGP. DGP 6 is simulated without a breakpoint, so the coverage rate is irrelevant and indicated by ×.

from 0.95 to 1, resulting in a state-wise non-stationary model. This change leads to breakpoint estimates very close to the true breakpoint, except in the Mean model. A similar outcome is observed in the fifth DGP, which features a larger increase in the AR-coefficient. In the sixth DGP, which is defined without any breakpoints, the Mean model estimates breakpoints near the midpoint of the sample period, while the other two specifications yield inconclusive results. In the seventh DGP, the AR and Fixed AR models produce estimates close to the true breakpoint. However, the Mean model continues to produce breakpoint estimates far from the true value. Examining the eighth DGP, the three models perform almost equally well.

Overall, the Fixed AR and AR models tend to perform well in non-stationary scenarios, estimating breakpoints close to the true breakpoints. The methodology, however, appears to struggle with accurately estimating the true breakpoint in cases of minor changes between states and large error term variance. In contrast, the Mean model does not perform well in DGPs featuring gradual changes, aligning with theoretical expectations as detailed in Bai and Perron (2003).

The coverage rate of a CI is the proportion of times the CI covers the true breakpoint, here at t=250. We find that the CIs of the Mean model are generally very wide and have varying coverage. In the Fixed AR and AR models, the CIs are typically narrower. The coverage rates are best in the DGPs with large differences between the states as seen in DGPs 4, 5, 7 and 8 using the Fixed AR model specification, which is in line with the findings of Bai and Perron (2003). For the AR model, the coverage rates are only close to the desired 95% in the seventh and eighth DGP, indicating that the CIs are inadequate in most of the DGPs considered.

Table C.3 shows the mean number of breakpoints estimated for each DGP and method, along with the proportion of correctly estimated breakpoints. The difficulty in accurately estimating gradual changes using the Mean model is also evident when estimating the number of breakpoints. This model specification leads to overestimating the number of breakpoints in all DGPs considered except DGP 8, where it performs well. The BIC criterion in the Fixed AR specification performs very well, with an estimated number of breakpoints equal to the true number in most simulations in DGP 2-8. The LWZ criterion performs almost equally well except in the third DGP, while the KT criterion vastly overestimates the number of breakpoints in DGP 1-7. In the AR model, the information criteria all perform well in DGPs 2-8 except for the third DGP where the LWZ criterion underestimates the number of breakpoints.

| DGP |           | Mean      |           |           | Fixed AR   |           | AR         |            |            |  |
|-----|-----------|-----------|-----------|-----------|------------|-----------|------------|------------|------------|--|
|     | BIC       | LWZ       | KT        | BIC       | LWZ        | KT        | BIC        | LWZ        | KT         |  |
| 1   | 3.0 (0%)  | 3.0 (0%)  | 3.0 (0%)  | 0.2 (15%) | 0.0 (0%)   | 3.0 (0%)  | 0.1 (6%)   | 0.0 (0%)   | 0.0 (3%)   |  |
| 2   | 3.0 (0%)  | 3.0 (0%)  | 3.0 (0%)  | 1.0 (97%) | 0.8 (82%)  | 3.0 (0%)  | 1.0 (94%)  | 0.5 (46%)  | 1.0 (93%)  |  |
| 3   | 2.9 (0%)  | 2.7 (4%)  | 3.0 (0%)  | 1.0 (94%) | 0.2 (16%)  | 2.9 (0%)  | 0.9 (85%)  | 0.0 (0%)   | 0.7 (70%)  |  |
| 4   | 3.0 (0%)  | 3.0 (0%)  | 3.0 (0%)  | 1.0 (98%) | 1.0 (98%)  | 2.8 (0%)  | 1.0 (99%)  | 0.9 (92%)  | 1.0 (99%)  |  |
| 5   | 3.0 (0%)  | 3.0 (0%)  | 3.0 (0%)  | 1.0 (99%) | 1.0 (97%)  | 2.7 (0%)  | 1.0 (99%)  | 1.0 (100%) | 1.0 (99%)  |  |
| 6   | 3.0 (0%)  | 3.0 (0%)  | 3.0 (0%)  | 0.0 (98%) | 0.0 (100%) | 3.0 (0%)  | 0.0 (100%) | 0.0 (100%) | 0.0 (100%) |  |
| 7   | 3.0 (0%)  | 3.0 (0%)  | 3.0 (0%)  | 1.0 (99%) | 1.0 (100%) | 3.0 (0%)  | 1.0 (98%)  | 1.0 (100%) | 1.0 (98%)  |  |
| 8   | 1.5 (63%) | 1.0 (98%) | 1.3 (72%) | 1.0 (99%) | 1.0 (100%) | 1.3 (73%) | 1.0 (100%) | 1.0 (98%)  | 1.0 (100%) |  |

**Table C.3.** Means of the estimated number of breakpoints for each model specification across different DGPs, rounded to one decimal. Percentages indicate the proportion of estimates equal to the true number of breakpoints.

Figure C.1. DGP 1: Left: Five process realizations. Right: The densities of the estimated breakpoints for each specification.

Figure C.2. DGP 2: Left: Five process realizations. Right: The densities of the estimated breakpoints for each specification.

Figure C.3. DGP 3: Left: Five process realizations. Right: The densities of the estimated breakpoints for each specification.

Figure C.4. DGP 4: Left: Five process realizations. Right: The densities of the estimated breakpoints for each specification.

Figure C.5. DGP 5: Left: Five process realizations. Right: The densities of the estimated breakpoints for each specification.

Figure C.6. DGP 6: Left: Five process realizations. Right: The densities of the estimated breakpoints for each specification.

Figure C.7. DGP 7: Left: Five process realizations. Right: The densities of the estimated breakpoints for each specification.

Figure C.8. DGP 8: Left: Five process realizations. Right: The densities of the estimated breakpoints for each specification.

#### C2 Serially correlated error term

A possible extension of the simulation study outlined in Eq. (C1) is allowing the error term to exhibit serial correlation. We use the same DGPs as before, but generate  $\{\varepsilon_t\}_{t=1}^T$  as follows,

$$\varepsilon_t = \psi \varepsilon_{t-1} + \theta \eta_{t-1} + \eta_t, \quad \eta_t \stackrel{i.i.d.}{\sim} \mathcal{N}(0, \sigma_n^2) \quad \forall t.$$
 (C2)

We conduct 1000 simulations for each, with a sample size of 500. Here, we consider DGPs 2, 3, 4, 5, 7, and 8 as outlined in Table C.1 and refer to these DGPs in the serially correlated cases as models  $2_s$ ,  $3_s$ ,  $4_s$ ,  $5_s$ ,  $7_s$ , and  $8_s$ . We set  $\psi = \theta = 0.5$  and the standard deviation  $\sigma_{\eta}$ , such that the standard deviation of  $\varepsilon_t$  corresponds to the  $\sigma$  in Table C.1. This is accomplished as follows,

465 
$$\operatorname{Var}(\varepsilon_{t}) = \operatorname{Var}(\psi \varepsilon_{t-1} + \theta \eta_{t-1} + \eta_{t})$$

$$= \psi^{2} \operatorname{Var}(\varepsilon_{t-1}) + \theta^{2} \operatorname{Var}(\eta_{t-1}) + 2\psi \theta \operatorname{Cov}(\varepsilon_{t-1}, \eta_{t-1}) + \operatorname{Var}(\eta_{t}).$$

$$= \psi^{2} \operatorname{Var}(\varepsilon_{t-1}) + \theta^{2} \sigma_{\eta}^{2} + 2\psi \theta \sigma_{\eta}^{2} + \sigma_{\eta}^{2},$$

since  $\varepsilon_{t-1}$  and  $\eta_{t-1}$  have zero means and  $\mathbb{E}[\varepsilon_t \eta_t] = \phi \mathbb{E}[\varepsilon_{t-1} \eta_t] + \theta \mathbb{E}[\eta_t \eta_{t-1}] + \mathbb{E}[\eta_t^2] = \sigma_\eta^2$ . Given stationarity of the process, which implies  $\sigma^2 = \text{Var}(\varepsilon_t)$  for all t, we derive,

$$\sigma_{\eta}^2 = \sigma^2 \frac{1 - \psi^2}{1 + \theta^2 + 2\psi\theta}.$$

470 This adjustment ensures the comparability of the results between the two error term types.

In Figs. C.9 through C.14, we plot examples of realizations and frequency plots of the estimated breakpoints using each of the models while imposing a single breakpoint in the estimation. The results are summarized in Table C.4, which provides means of the estimated breakpoints and medians of the lower and upper boundary of the estimated confidence intervals, along with the coverage rates for each model specification and DGP. Generally speaking, the mean of the estimated breakpoints are further from the true breakpoint and the CIs become wider compared to the results from the corresponding DGPs without serial correlation. It is evident that serial correlation in the error term makes it more difficult to estimate the dating of breaks. We find that the Fixed AR and AR models perform well for DGP  $7_s$ , which has a large difference between the states and low variance. This is in line with the theoretical framework by Bai and Perron (2003), who note that the estimated break dates are consistent even in the presence of serial correlation. The Fixed AR model performs well in DGPs  $2_s$ ,  $4_s$  and  $5_s$  where the mean of the estimated breakpoints is close to the true breakpoint, and confidence intervals are reasonably wide with acceptable coverage rates. The results of the AR model are less conclusive.

For the Mean and Fixed AR models, the coverage rates are generally close to the desired 95% and even higher in some DGPs. However, the CIs are also extremely wide, reaching outside the sample window in many DGPs. The CIs seem reasonable in the Fixed AR model for DGPs  $2_s$ ,  $4_s$ ,  $5_s$ , and  $7_s$ , where the coverage rates are close to 95% and the medians of the lower and upper bounds of the CIs are not too extreme. The CIs for the AR model are generally wider than in the version without serial correlation in the error term. In the AR model, the coverage rates are lower than the desired 95%, but it seems that DGPs with large breaks have higher coverage rates. The relatively poor performance is in line with the theoretical framework by Bai and Perron (2003). The authors note that the construction of the CIs rely on having no serial correlation in the error term if a lagged dependent variable is included as a regressor that has coefficients that are subject to breakpoints.

Table C.5 shows the mean number of breakpoints estimated for each DGP and method, along with the proportion of correctly estimated number. In the Mean model, all information criteria overestimate the number of breakpoints. An important exception is the eighth DGP,

| DGP   |         | N     | Mean  |          | Fixed AR |       |       |          | AR      |       |       |          |
|-------|---------|-------|-------|----------|----------|-------|-------|----------|---------|-------|-------|----------|
|       | BP est. | Lower | Upper | Coverage | BP est.  | Lower | Upper | Coverage | BP est. | Lower | Upper | Coverage |
| $2_s$ | 332     | -1400 | 335   | 95.9%    | 247      | 188   | 312   | 95.7%    | 261     | 190   | 299   | 79.9%    |
| $3_s$ | 266     | 60    | 787   | 90.6%    | 285      | -112  | 656   | 97.2%    | 276     | 156   | 421   | 77.1%    |
| $4_s$ | 340     | -776  | 339   | 94.9%    | 252      | 197   | 301   | 96.9%    | 264     | 195   | 277   | 84.9%    |
| $5_s$ | 342     | -329  | 340   | 96.2%    | 256      | 196   | 266   | 96.4%    | 259     | 192   | 250   | 70.8%    |
| $7_s$ | 333     | -1708 | 329   | 92.3%    | 249      | 230   | 270   | 97.6%    | 251     | 230   | 267   | 92.8%    |
| $8_s$ | 250     | 122   | 370   | 98.3%    | 245      | -5    | 492   | 99.8%    | 247     | 23    | 490   | 97.4%    |

**Table C.4.** Mean of the estimated breakpoints and medians of the lower and upper boundary of the estimated confidence intervals, along with the coverage rates for each model specification and DGP.

where the performance is better, as in the case without serial correlation. In the Fixed AR and AR model specifications, the LWZ criterion generally performs well, while both the BIC and the KT criteria generally overestimate. However, the LWZ criterion leads to underestimating the number of breakpoints in DGPs  $3_s$  and  $8_s$ . These two DGPs are characterized by fixed AR-coefficients that are lower than one. This implies that these two processes do not exhibit an autoregressive unit root. Hence, it seems that the LWZ criterion performs well in cases of state-wise non-stationarity or switching between stationary and non-stationary states.

495

Compared to the findings in the DGPs without serial correlation, it is clear that the proportion of correct estimates are lower for most DGPs and model specifications. Overall, the best performing criterion seems to be the LWZ criterion in the Fixed AR and AR models, while the Mean model typically leads to overestimating the number of breakpoints.

| DGP   |           | Mean      |           |           | Fixed AR  |           | AR        |           |           |  |
|-------|-----------|-----------|-----------|-----------|-----------|-----------|-----------|-----------|-----------|--|
|       | BIC       | LWZ       | KT        | BIC       | LWZ       | KT        | BIC       | LWZ       | KT        |  |
| $2_s$ | 3.0 (0%)  | 3.0 (0%)  | 3.0 (0%)  | 1.9 (32%) | 0.9 (70%) | 2.9 (0%)  | 1.8 (37%) | 0.7 (61%) | 1.9 (33%) |  |
| $3_s$ | 3.0 (0%)  | 2.8 (2%)  | 3.0 (0%)  | 0.7 (33%) | 0.0 (0%)  | 2.7 (3%)  | 0.3 (19%) | 0.0 (0%)  | 0.4 (17%) |  |
| $4_s$ | 3.0 (0%)  | 3.0 (0%)  | 3.0 (0%)  | 1.7 (45%) | 1.0 (85%) | 2.8 (1%)  | 1.6 (51%) | 0.8 (79%) | 1.6 (47%) |  |
| $5_s$ | 3.0 (0%)  | 3.0 (0%)  | 3.0 (0%)  | 1.8 (5%)  | 1.1 (85%) | 2.8 (0%)  | 1.7 (40%) | 1.0 (92%) | 1.6 (49%) |  |
| $7_s$ | 3.0 (0%)  | 3.0 (0%)  | 3.0 (0%)  | 1.9 (34%) | 1.1 (89%) | 3.0 (0%)  | 1.9 (34%) | 1.0 (96%) | 1.9 (32%) |  |
| 8.    | 2.2 (21%) | 1.2 (78%) | 2.2 (23%) | 0.4 (35%) | 0.0 (0%)  | 1.9 (36%) | 0.0 (4%)  | 0.0 (0%)  | 0.0 (3%)  |  |

**Table C.5.** Means of the estimated number of breakpoints for each model specification across different DGPs, rounded to one decimal. Percentages indicate the proportion of estimates equal to the true number of breakpoints.

Figure C.9. DGP  $2_s$ : Left: Five process realizations. Right: The densities of the estimated breakpoints for each specification.

Figure C.10. DGP 3<sub>s</sub>: Left: Five process realizations. Right: The densities of the estimated breakpoints for each specification.

Figure C.11. DGP  $4_s$ : Left: Five process realizations. Right: The densities of the estimated breakpoints for each specification.

Figure C.12. DGP 5<sub>s</sub>: Left: Five process realizations. Right: The densities of the estimated breakpoints for each specification.

Figure C.13. DGP 7<sub>s</sub>: Left: Five process realizations. Right: The densities of the estimated breakpoints for each specification.

Figure C.14. DGP 8<sub>s</sub>: Left: Five process realizations. Right: The densities of the estimated breakpoints for each specification.

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
