# Peer review of "Estimating breakpoints in the Cenozoic Era: An econometric approach"

_EGUsphere, 2024_

## Author Comment (AC1)

**Responses to Referee 1**

**General Comments:**

This paper is a useful contribution to paleoclimate literature. The authors present the usage of a novel breakpoint analysis technique on a record of Cenozoic climate from Westerhold et al. (2020), and adequately support the usage of this technique on said data via sensitivity tests.

My major piece of feedback is that this paper currently lacks substantive earth science/paleoclimate motivation, and is far too terse. While it's an impressive piece of work on the application of this changepoint technique to CENOGRID, additional context is necessary for broader application in the paleosciences. That is, as a methods paper that is designed to introduce a new approach to changepoint analysis within the paleosciences, further work is necessary to show where and when this method can be used to answer other paleoclimate questions. For example, some discussion of age uncertainty is essential, as this is an issue of fundamental importance in paleoclimate. Additionally, CENOGRID is an interesting dataset, but its length and completeness aren't typical of paleoclimate records, which complicates its being the sole non-synthetic example used in demonstrating the application of a novel technique. Additional explanation of the choice of CENOGRID as well as potential edge cases not covered by CENOGRID needs to be done. As a data scientist I'm left feeling confident that this technique is useful, and I'm intrigued by the idea of applying this method to my own work. However, as an earth scientist I'm left not quite understanding the breadth of problems that it is well suited for, nor what the significance of the additional breakpoints that were identified in the Cenozoic is.

We appreciate the reviewer's recognition of our work as a useful contribution to the paleoclimate literature. The reviewer raises an important point about strengthening the paleoclimate motivation and further demonstrating the broad applicability of our method. We acknowledge that the manuscript primarily focuses on the technical aspects of our approach, and that its relevance within the paleosciences can be better contextualized. To address this, we will expand our discussion to clarify where and when this technique can be applied to other paleoclimate problems. Specifically, we will:

- Strengthen the paleoclimate motivation and terminology.

- Discuss specific limitations of CENOGRID and provide a more detailed justification for its selection as the primary dataset.

Furthermore, we can include an additional application to further demonstrate the breadth of problems where the methodology can be applied:

- A shorter time series of $\delta^{18}O$ from Greenland ice cores (Seierstad et al., 2014), where our approach allows for the detection of breakpoints while including orbital factors (eccentricity, obliquity, and precession) as explanatory variables. This application will illustrate the method's utility in detecting transitions associated with Dansgaard-Oeschger events also considered by Livina et al. (2010). However, we plan to include this in an appendix to keep the focus on the CENOGRID in the main text. See preliminary findings in Section 1.

**Specific Comments:**

The need for this method in particular within paleoclimatology should be discussed in more detail. This paragraph *"Our approach contributes to the existing breakpoint detection methods in paleoclimate research by applying well-established econometric tools in the time-domain, developed in Bai and Perron (1998, 2003), to identify climate states in the paleo record. It enables the estimation of multiple breakpoints along with confidence intervals and provides procedures to estimate the number of breakpoints"* should be built upon. I see how confidence intervals might be useful, but what are the other strengths and weaknesses of this approach when compared to other methods? Some discussion of other approaches is offered in the preceding paragraph, but it is somewhat superficial. As a reader, I need better context for breakpoint analysis in paleoclimate studies: its historical usage, current applications, and future potential.

We will strengthen our discussion of our method's advantages and positioning within existing breakpoint detection techniques in paleoclimate research. Specifically, we will do the following:

- Provide a concise historical overview of breakpoint analysis in paleoclimatology, summarizing previous methods and applications.

- Clarify why the Bai and Perron (1998, 2003) approach is well-suited for paleoclimate applications.

- Highlight the key advantages of our approach and compare it with the existing methods in the paleoclimate literature. Also, we will emphasize the possibility of including explanatory variables in our approach.

- Acknowledge limitations of our approach, including assumptions of piecewise linearity, computational requirements, and the handling of irregularly spaced time series.

- We will also include a discussion on future applications and possible extensions.

This sentence "*The paleoclimate variable $\delta 18O$ measures the ratio of 18O to 16O in the shells of benthic foraminifera obtained from ocean sediment cores, relative to a standard sample.*" is incorrect (or at least misleading), as $\delta 18O$ is not exclusive to benthic forams, which the sentence seems to be suggesting.

We thank the reviewer for spotting this. We will correct the statement to avoid confusion and ensure accuracy.

This sentence "*The weight difference between the oxygen isotopes leads to an inverse relationship between $\delta 18O$ and ocean temperatures; see for instance Epstein et al. (1951) and Shackleton (1967).*" is an inadequate description of benthic $\delta 18O$. Mention of other factors (seawater composition, ice sheet volume, etc.) needs to be included. CENOGRID is composed of many integrated signals, the makeup of which will determine what detected breakpoints are telling us about the climate.

We will revise the description of benthic $\delta^{18}O$ to include additional factors such as seawater composition and ice volume effects. Also, we will clarify that when we write $\delta^{18}O$, we are referring to benthic $\delta^{18}O$.

As the reader, I'm left wondering why only oxygen isotopes were considered. Carbon isotopes are also available, why not include carbon isotopes in the analysis, as was done in the original Westerhold publication?

Our method is flexible and can certainly be applied to other time series, including carbon isotopes from Westerhold et al. (2020). However, to better illustrate the broad applicability of our approach, we can include an additional analysis of oxygen isotopes from ice cores. This example highlights how the method can be used in different paleoclimate contexts and demonstrates how explanatory variables, such as orbital factors, can be seamlessly incorporated into the framework. We will consider having this additional illustration in an appendix.

The authors discuss the varying resolution of the time series at length, which is helpful. However, I would be curious as to whether or not the resolution impacted the detection of break points. Is there any correlation between the detection of new breakpoints (discussed later in the manuscript) and the resolution of the time series? For example, just by visual comparison, it seems to me that the breakpoint observed in Coolhouse 1 in later sections might be related to a large change in resolution that occurs nearby. While this breakpoint isn't heavily interpreted here, this is an important point to understand if other researchers are to apply this method to their own data.

The resolution issue essentially reflects a bias-variance trade-off, where a higher binning frequency (i.e., higher resolution) reduces variance of the breakpoints but may introduce some bias, whereas a lower binning frequency is more likely to increase variance. This means that the primary effect of changing the resolution is on the constructed confidence intervals rather than on the location of the breakpoints themselves. This is evident in Figure 3 of the manuscript, where a lower binning frequency generally leads to wider confidence intervals. Additionally, we observe strong stability in the estimated breakpoints across different binning frequencies, suggesting that resolution does not systematically impact breakpoint detection.

Regarding the estimated breakpoint in Coolhouse 1, it is indeed correct that this breakpoint is fairly close to a change in the resolution of the data. We agree that binned data reflect the quality of the original dataset, which can make it more challenging to accurately estimate breakpoints near or within data gaps. However, from a visual perspective, the breakpoint in Coolhouse 1 could also be influenced by the hump-shaped pattern that follows shortly after, between 17 and 14 million years ago. We plan to add a more insightful discussion on this issue.

This sentence "*Furthermore, we recommend using binning frequencies 10 and 25 kyr as they result in the most consistent outcomes.*" strikes me as rigid and somewhat unhelpful, as many records will not share the general time axis properties of CENOGRID. Is there a different way to describe your binning recommendation that's more flexible and/or applicable to other datasets?

We acknowledge the referee's concern and will clarify that the choice of binning frequency should be tailored to the characteristics of the time series at hand. We will recommend selecting a frequency that maintains sufficient observations per bin while considering the dataset's length and resolution. If the time series is already equidistant, we suggest keeping the original resolution. We will revise the sentence to specify that our recommendation applies to this particular dataset and provide guidelines for other applications separately.

Certain technical choices need to be better explained given the audience of this journal. For example: "*To address these issues, we use the autocorrelation and heteroscedasticity consistent (HAC) covariance matrix estimator with prewhitening in our implementations.*". Perhaps this is standard fare in breakpoint analysis literature, but most paleoclimatologists won't be familiar with this procedure. Some explanation as to why this approach is suitable for this data is warranted here, as is mention of alternatives that were considered. In the same vein, it would be helpful to spend a little bit more time explaining information criteria. That is, expand upon "We use information criteria to estimate the number of breakpoints". What does this mean, why are they used, have they been used in paleoclimate contexts before, etc.

We appreciate the referee's feedback and agree that additional explanation is needed to make these methodological choices more accessible to a paleoclimate audience. In the revised manuscript, we will expand on the rationale for using the HAC covariance matrix estimator with prewhitening, explaining its role in addressing autocorrelation and heteroscedasticity, which are common in paleoclimate time series. We will also briefly discuss why our chosen approach is particularly suited for this application. The presence of autocorrelation and heteroscedasticity in paleoclimate time series has been considered in previous work focusing on ice core records, including Davidson et al. (2015), who considers heteroscedasticity, and Keyes et al. (2023), who investigate autocorrelation. In contrast, our manuscript primarily extends focus on paleoclimate records from ocean sediment cores, where similar challenges arise.

We will also clarify the use of information criteria for breakpoint estimation, explaining their role in model selection and previous applications in paleoclimate studies. Furthermore, we will include the $R^2$ as a complementary goodness-of-fit measure, to enhance accessibility for a broader readership.

Age uncertainty needs to be addressed somewhere in this paper. It doesn't need a full treatment, in that the method doesn't need to be modified to account for it, nor does it need to be included in the analysis, but discussion of how to include it in future studies is essential. Specifically I would be curious as to how this technique might be expanded to include the usage of age ensembles and how choice of age modeling method might impact breakpoint detection. However, discussion of including age uncertainty directly would also be acceptable here.

We will include a brief discussion on age uncertainty and age ensembles. While this is an important issue, we are not aware of any methodology in the paleoclimate literature that

explicitly treats a time stamp as a random variable, with its variance representing the uncertainty of the time stamp. This challenge has also not been addressed within our framework, and incorporating the time-stamp uncertainty would require significant methodological developments. Such an extension involves advanced statistical techniques that are beyond the scope of this study.

That said, addressing age uncertainty is a crucial direction for future research. We will highlight this in the manuscript and reference relevant studies that discuss the issue (Telford et al., 2004; Franke and Donner, 2019), as well as papers that explore aspects of transition detection in the presence of age uncertainty, such as Goswami et al. (2018).

The simulation study is a particular strength of this work. The authors thoroughly test their method across different data-generating processes, demonstrating its robustness to various forms of non-stationarity and serial correlation. This kind of rigorous testing is essential for establishing the reliability of statistical methods in paleoclimate contexts. I just wanted to make a note of that.

We appreciate this positive feedback.

When analyzing the possible presence of multiple breakpoints, I'm left desiring some kind of prescription as to how I should set the number of breakpoints. Certainly the claim that there are more than 5 statistically significant breakpoints in CENOGRID seems robust. However, the current analysis feels somewhat hand-wavy, with seven breakpoints being settled upon in a rather arbitrary way. In particular this statement needs to be expounded upon: "*The estimation results based on information criteria justify dividing the climate states Warmhouse II and Coolhouse II into two substates each at approximately 39.7 Ma and 10 Ma, respectively. This is supported by the presence of breakpoints estimated approximately at these time stamps in the estimations with seven or more breakpoints.*"

We recognize the need for a clearer justification of the chosen number of breakpoints. In the revised manuscript, we will provide a more detailed explanation of how information criteria guide breakpoint selection and why seven breakpoints were ultimately chosen. Specifically, we will elaborate on the evidence supporting the subdivision of Warmhouse II and Coolhouse II, highlighting how breakpoints at approximately 39.7 Ma and 10 Ma consistently emerge in models with seven or more breakpoints.

The ending of this manuscript is far too abrupt. Potentially new breakpoints are discovered

when varying numbers of breakpoints are allowed, but what do they mean? A few climate events are referenced, but events themselves may or may not justify entirely new regimes. Much context is needed here, interpreting and explaining the presence of these novel breakpoints. While the authors are free to choose how to address this comment, I might suggest including a "Discussion" section, in which the primary results are emphasized, and an explanation/interpretation of these results is offered. Some of my other comments could probably be folded into this section as well.

We appreciate this suggestion and agree that a more structured discussion will improve the manuscript. As time series analysts, we do not claim to have the expertise to define entirely new climate regimes in the Cenozoic Era. However, from a statistical perspective, our results strongly indicate that these time points mark shifts in the underlying dynamics of the time series, distinguishing them from other periods.

To address this, we will add a "Discussion" section where we highlight and interpret our primary findings and their implications for paleoclimate time series analysis and for the understanding of the Cenozoic climate. This section will also include a brief discussion of age uncertainty and the limitations of the CENOGRID dataset. Additionally, we will provide insights for other researchers working with similar data, ensuring that our methodological contributions are framed within a broader paleoclimate context.

**Technical Comments:**

The paper currently is a bit undercited. I suggest the authors go back through with a fine toothed comb and make sure they're citing existing literature wherever possible. In particular, all sections discussing $\delta 18O$ interpretation should be thoroughly cited, particularly regarding ice volume effects, temperature relationships, etc. A couple of other key spots (non-exhaustive) that need citations include:

- *"The climatic transitions contain important information about variations in Earth's climate system"* (here Tierney et al. 2020 is referenced, but some explanation of what is contained in that review along with additional citations is called for)

- *"Our approach contributes to the existing breakpoint detection methods in paleoclimate research"* (cite breakpoint analysis in paleoclimate literature)

- *"This breakpoint aligns with the Middle Eocene Climatic Optimum, a known climatic event"* (cite original papers describing this event)

- "Some of these breakpoints coincide with other climatic events, for instance, the Latest Danian Event at 62.2 Ma and the onset of the Miocene Climatic Optimum at 16.9 Ma" (cite original papers describing these events, not just Westerhold 2020)

Thank you for this detailed feedback. We will carefully review the manuscript to ensure that the relevant literature is cited appropriately. In particular, we will strengthen the citations in sections discussing $\delta^{18}O$ interpretation, providing references on ice volume effects, temperature relationships, and other key aspects. Here, we will acknowledge that benthic $\delta^{18}O$ reflects of many signals including deep-ocean temperatures, ice volume, and sea-water salinity. As a citation, we will consider Waelbroeck et al. (2002) who provided a comprehensive analysis of sea-level and deep-ocean temperature changes derived from benthic $\delta^{18}O$ records, and Oerlemans (2004), who proposed corrections to the Cenozoic $\delta^{18}O$ deep-sea temperature record to account for Antarctic ice volume. For a detailed discussion on disentangling these signals, we will refer to Berends et al. (2021). However, we will also emphasize that our goal is not to disentangle the individual signals embedded in the record, but rather to estimate climate states that reflect the aggregated climate signals.

Furthermore, we will ensure that:

- The discussion of climatic transitions includes a more detailed explanation of Tierney et al. (2020) along with more citations speaking to the importance of Cenozoic climate states. As argued by Burke et al. (2018), we will note that the Cenozoic climate states are valuable for finding analogs for modern warming scenarios. Also, we will mention that several studies find that the climate sensitivity is state-dependent (Caballero and Huber, 2013) and for this the Cenozoic climate states play an important role. Furthermore, it will be noted that planktonic foraminifera records show that Cenozoic climate shifts have greatly influenced marine ecosystems (Swain et al., 2024).

- Our contribution to breakpoint detection in paleoclimate research is contextualized by referencing the review studies of Mudelsee et al. (2014) on benthic $\delta^{18}O$ time series analysis in the Cenozoic Era and Marwan et al. (2021) on nonlinear time series analysis of paleoclimate data, along with relevant applications discussed in these reviews.

- Statements about specific climatic events, such as the Middle Eocene Climatic Optimum, the Latest Danian Event, and the Miocene Climatic Optimum, are supported by original studies rather than secondary sources. Specifically, we will cite Bohaty and Zachos (2003) for the Middle Eocene Climatic Optimum, Bornemann et al. (2009) for the Latest Danian

Event, and Flower and Kennett (1994) and Zachos et al. (2001) for the Miocene Climatic Optimum.

**1 $\delta^{18}$O from ice cores extracted in Greenland**

This section shows some preliminary results for estimation of breakpoints in an ice core record. We consider the NGRIP dataset (Seierstad et al., 2014) and aim to estimate Dansgaard-Oeschger events. This is inspired by Livina et al. (2010) who use potential analysis to determine the number of breakpoints in a paleoclimate time series of $\delta^{18}$O stemming from ice cores from Greenland.

**Context:**

- The dataset consists of Greenland ice-core $\delta^{18}$O data from Seierstad et al. (2014), an updated version of the time series analyzed by Livina et al. (2010). The dataset spans from 60,000 years before present to today. The Last Ice Age lasted from approximately 115,000 to 11,700 years ago.

- Dansgaard-Oeschger (D-O) events are rapid warming periods during the Last Ice Age (Dansgaard et al., 1993). The vertical lines in the plots on the next page indicate the four D-O events mentioned by Livina et al. (2010). There are of course more D-O events, which we will consider in future versions of this analysis.

We aim to detect D-O events using breakpoint detection with the same AR model specification as before.

- First, we estimate breakpoints and confidence intervals without accounting for orbital factors. The results are shown in Figure 1 and indicate that the estimated breakpoints align well with known D-O events.

- We are then incorporating orbital factors, namely eccentricity ($E_t$), obliquity ($O_t$), and precession index ($Pr_t$) when detecting the breakpoints. The results are shown in Figure 2, where we observe similar breakpoints as in the previous case, but with slightly better alignment to the D-O events.

[Figure]

Figure 1: Estimated breakpoints using the AR model for one to six breakpoints on the NGRIP data. Black ×'s indicate estimated breakpoints, while colored shaded rectangles represent 95% confidence intervals. The results overlay the $\delta^{18}O$ dataset from Seierstad et al. (2014), with vertical dashed lines marking the D-O events considered in Livina et al. (2010).

[Figure]

Figure 2: stimated breakpoints using the AR model for one to six breakpoints on the NGRIP data, while including **orbital factors** in the model. Black ×'s indicate estimated breakpoints, while colored shaded rectangles represent 95% confidence intervals. The results overlay the $\delta^{18}O$ dataset from Seierstad et al. (2014), with vertical dashed lines marking the D-O events considered in Livina et al. (2010).

**References**

Berends, C. J., de Boer, B., and van de Wal, R. S. W. (2021). Reconstructing the evolution of ice sheets, sea level, and atmospheric $CO_2$ during the past 3.6 million years. *Climate of the Past*, 17(1):361–377.

Bohaty, S. M. and Zachos, J. C. (2003). Significant Southern Ocean warming event in the late middle Eocene. *Geology*, 31(11):1017–1020.

Bornemann, A., Schulte, P., Sprong, J., Steurbaut, E., Youssef, M., and Speijer, R. P. (2009). Latest Danian carbon isotope anomaly and associated environmental change in the southern Tethys (Nile Basin, Egypt). *Journal of the Geological Society*, 166(6):1135–1142.

Burke, K. D., Williams, J. W., Chandler, M. A., Haywood, A. M., Lunt, D. J., and Otto-Bliesner, B. L. (2018). Pliocene and eocene provide best analogs for near-future climates. *Proceedings of the National Academy of Sciences*, 115(52):13288–13293.

Caballero, R. and Huber, M. (2013). State-dependent climate sensitivity in past warm climates and its implications for future climate projections. *Proceedings of the National Academy of Sciences*, 110(35):14162–14167.

Dansgaard, W., Johnsen, S., Clausen, H., Dahl-Jensen, D., Gundestrup, N., Hammer, C., Hvidberg, C., Steffensen, J., Sveinbjörnsdottir, A., Jouzel, J., and GC, B. (1993). Evidence of general instability of past climate from a 250-kyr ice-core record. *Nature*, 364:218–220.

Davidson, J., Stephenson, D., and Turasie, A. (2015). Time series modeling of paleoclimate data. *Environmetrics*, 27:n/a–n/a.

Flower, B. P. and Kennett, J. P. (1994). The middle Miocene climatic transition: East Antarctic ice sheet development, deep ocean circulation and global carbon cycling. *Palaeogeography, Palaeoclimatology, Palaeoecology*, 108:537–555.

Franke, J. G. and Donner, R. V. (2019). Correlating paleoclimate time series: Sources of uncertainty and potential pitfalls. *Quaternary Science Reviews*, 212:69–79.

Goswami, B., Boers, N., Rheinwalt, A., Marwan, N., Heitzig, J., Breitenbach, S., and Kurths, J. (2018). Abrupt transitions in time series with uncertainties. *Nature Communications*, 9(48).

Keyes, N. D. B., Giorgini, L. T., and Wettlaufer, J. S. (2023). Stochastic paleoclimatology: Modeling the EPICA ice core climate records. *Chaos*, 33(9):093132. Special Collection: Theory-informed and Data-driven Approaches to Advance Climate Sciences.

Livina, V. N., Kwasniok, F., and Lenton, T. M. (2010). Potential analysis reveals changing number of climate states during the last 60 kyr. *Climate of the Past*, 6(1):77–82.

Marwan, N., Donges, J. F., Donner, R. V., and Eroglu, D. (2021). Nonlinear time series analysis of palaeoclimate proxy records. *Quaternary Science Reviews*, 274:107245.

Mudelsee, M., Bickert, T., Lear, C. H., and Lohmann, G. (2014). Cenozoic climate changes: A review based on time series analysis of marine benthic $\delta$18O records. *Reviews of Geophysics*, 52(3):333–374.

Oerlemans, J. (2004). Correcting the Cenozoic $\delta^{18}$O deep-sea temperature record for Antarctic ice volume. *Palaeogeography, Palaeoclimatology, Palaeoecology*, 208(3):195–205.

Seierstad, I. K., Abbott, P. M., Bigler, M., Blunier, T., Bourne, A. J., Brook, E., Buchardt, S. L., Buizert, C., Clausen, H. B., Cook, E., Dahl-Jensen, D., Davies, S. M., Guillevic, M., Johnsen, S. J., Pedersen, D. S., Popp, T. J., Rasmussen, S. O., Severinghaus, J. P., Svensson, A., and Vinther, B. M. (2014). Consistently dated records from the Greenland GRIP, GISP2 and NGRIP ice cores for the past 104 ka reveal regional millennial-scale 18O gradients with possible Heinrich event imprint. *Quaternary Science Reviews*, 106:29–46. Dating, Synthesis, and Interpretation of Palaeoclimatic Records and Model-data Integration: Advances of the INTIMATE project(INTegration of Ice core, Marine and TErrestrial records, COST Action ES0907).

Swain, A., Woodhouse, A., Fagan, W. F., Fraass, A. J., and Lowery, C. M. (2024). Biogeographic response of marine plankton to cenozoic environmental changes. *Nature*.

Telford, R., Heegaard, E., and Birks, H. (2004). All age–depth models are wrong: but how badly? *Quaternary Science Reviews*, 23(1-2):1–5.

Tierney, J. E., Poulsen, C. J., Montañez, I. P., Bhattacharya, T., Feng, R., Ford, H. L., Hönisch, B., Inglis, G. N., Petersen, S. V., Sagoo, N., Tabor, C. R., Thirumalai, K., Zhu, J., Burls, N. J., Foster, G. L., Goddéris, Y., Huber, B. T., Ivany, L. C., Turner, S. K., Lunt, D. J., McElwain, J. C., Mills, B. J. W., Otto-Bliesner, B. L., Ridgwell, A., and Zhang, Y. G. (2020). Past climates inform our future. *Science*, 370(6517):eaay3701.

Waelbroeck, C., Labeyrie, L., Michel, E., Duplessy, J. C., McManus, J. F., Lambeck, K., Balbon, E., and Labracherie, M. (2002). Sea-level and deep water temperature changes derived from benthic foraminifera isotopic records. *Quaternary Science Reviews*, 21(1–3):295–305.

Westerhold, T., Marwan, N., Drury, A. J., Liebrand, D., Agnini, C., Anagnostou, E., Barnet, J. S. K., Bohaty, S. M., Vleeschouwer, D. D., Florindo, F., Frederichs, T., Hodell, D. A., Holbourn, A. E., Kroon, D., Lauretano, V., Littler, K., Lourens, L. J., Lyle, M., Pälike, H., Röhl, U., Tian, J., Wilkens, R. H., Wilson, P. A., and Zachos, J. C. (2020). An astronomically dated record of Earth's climate and its predictability over the last 66 million years. *Science*, 369(6509):1383–1387.

Zachos, J., MO, P., Sloan, L., Thomas, E., and Billups, K. (2001). Trends, Rhythms, and Aberrations in Global Climate 65 Ma to Present. *Science (New York, N.Y.)*, 292:686–93.

---

## Author Response (AR1)

**Author's response**

**April, 2025**

**1 Editor:**

"The manuscript has been carefully evaluated by two reviewers with a background in this field. Both recognize the value of exploring new methodologies for delineating climate states, yet both express the sentiment that the contribution needs to better detail how the breakpoint detection methodology enhances our knowledge of Cenozoic climate. That is, in addition to "Strengthen[ing] the paleoclimate motivation and terminology", as the authors mention in their response, it is also imperative that they discuss the paleoclimatic consequences of their results, comparing them to those obtained from different methods, and discussing the significance of the similarities or differences that arise. As suggested in the response, the idea of applying this method to ice core data from Greenland is also valuable, though will probably require a change of focus/title, and will also need to discuss the paleoclimate implications of the results.

In summary, though there is potential value in applying this method to paleoclimate problems, in this journal the latter take prominence - not the method itself. As such, it may be useful for the authors to team up with or more co-authors with expertise on these questions, to make sure the work is a durable contribution to the paleoclimate literature.

The reviewers further provide very detailed suggestions for more specific improvements."

We appreciate the constructive comments and suggestions. As suggested, we have collaborated with an additional co-author with expertise in paleoclimatology. We believe that this collaboration has significantly improved the paleoclimate interpretations, terminology, and discussions throughout the manuscript. In the revised version, we have strengthened the paleoclimate motivation and expanded the discussion of our findings' implications across the introduction, data description, discussion, and conclusion sections. While a portion of the paper remains focused on methodological development and its broader applicability, as encouraged by Reviewer 1, the paleoclimatic context has been considerably expanded and deepened. We have also provided a more detailed comparison with other methods previously applied to the same record and discussed the similarities and differences with these established techniques.

Regarding the suggestion of applying the method to Greenland ice core data, we agree that this would require a major shift in focus and title. After careful consideration, we decided not to pursue this in the current manuscript. Instead, we incorporated an additional analysis focused on the Icehouse period using the Westerhold dataset. This revealed breakpoints close to the Mid-Pleistocene Transition (MPT) and the onset of Northern Hemisphere Glaciation, further demonstrating the applicability of our method in shorter samples. Accordingly, we have updated the title to better reflect the broader scope, and to acknowledge the methodological origin. We have also introduced the name "Bai-Perron" for the framework to ease future referencing.

**2 Referee 1:**

**General Comments:**

"This paper is a useful contribution to paleoclimate literature. The authors present the usage of a novel breakpoint analysis technique on a record of Cenozoic climate from Westerhold et al. (2020), and adequately support the usage of this technique on said data via sensitivity tests.

My major piece of feedback is that this paper currently lacks substantive earth science/paleoclimate motivation, and is far too terse. While it's an impressive piece of work on the application of this changepoint technique to CENOGRID, additional context is necessary for broader application in the paleosciences. That is, as a methods paper that is designed to introduce a new approach to changepoint analysis within the paleosciences, further work is necessary to show where and when this method can be used to answer other paleoclimate questions. For example, some discussion of age uncertainty is essential, as this is an issue of fundamental importance in paleoclimate. Additionally, CENOGRID is an interesting dataset, but its length and completeness aren't typical of paleoclimate records, which complicates its being the sole non-synthetic example used in demonstrating the application of a novel technique. Additional explanation of the choice of CENOGRID as well as potential edge cases not covered by CENOGRID needs to be done. As a data scientist I'm left feeling confident that this technique is useful, and I'm intriqued by the idea of applying this method to my own work. However, as an earth scientist I'm left not quite understanding the breadth of problems that it is well suited for, nor what the significance of the additional breakpoints that were identified in the Cenozoic is." We have strengthened the paleoclimate motivation and provided broader context for the methodology throughout the paper. Specifically:

- We expanded the paleoclimate motivation and terminology across the manuscript.
- We justified the choice of the CENOGRID dataset and discussed its limitations in more detail (Introduction and Section 3.3).
- We added an additional analysis focusing on the Icehouse period to demonstrate broader applicability (Section 3.2).

• We expanded the conclusion to outline potential future applications.

**Specific Comments:**

"The need for this method in particular within paleoclimatology should be discussed in more detail. This paragraph 'Our approach contributes to the existing breakpoint detection methods in paleoclimate research by applying well-established econometric tools in the time-domain, developed in Bai and Perron (1998, 2003), to identify climate states in the paleo record. It enables the estimation of multiple breakpoints along with confidence intervals and provides procedures to estimate the number of breakpoints' should be built upon. I see how confidence intervals might be useful, but what are the other strengths and weaknesses of this approach when compared to other methods? Some discussion of other approaches is offered in the preceding paragraph, but it is somewhat superficial. As a reader, I need better context for breakpoint analysis in paleoclimate studies: its historical usage, current applications, and future potential."

We expanded the discussion of our method's advantages and positioning within existing breakpoint detection methods:

- We include a concise overview of breakpoint analysis in paleoclimatology (Introduction).
- We have built on the mentioned sentence and related our approach to established ones more clearly. (Introduction)
- We provide clear explanation of the strengths and limitations of the Bai-Perron framework (Introduction, Discussion, Conclusion).
- We put emphasis on the flexibility of the framework, which for instance allows for including explanatory variables (Conclusion).
- We have added a section dedicated to discussing the limitations of our framework including modelling assumptions, computational requirements, age uncertainty, and irregular spacing challenges. (Section 3.3)

"This sentence 'The paleoclimate variable  $\delta 180$  measures the ratio of 18O to 16O in the shells of benthic foraminifera obtained from ocean sediment cores, relative to a standard sample.' is incorrect (or at least misleading), as  $\delta 180$  is not exclusive to benthic forams, which the sentence seems to be suggesting."

We thank the reviewer for spotting this. We have corrected the statement to ensure accuracy.

"This sentence 'The weight difference between the oxygen isotopes leads to an inverse relationship between  $\delta 180$  and ocean temperatures; see for instance Epstein et al. (1951) and Shackleton (1967).' is an inadequate description of benthic  $\delta 180$ . Mention of other factors (seawater composition, ice

sheet volume, etc.) needs to be included. CENOGRID is composed of many integrated signals, the makeup of which will determine what detected breakpoints are telling us about the climate."

We have revised the description of benthic  $\delta^{18}O$  to include additional factors such as seawater composition and ice volume effects. Also, we have clarified that when we write  $\delta^{18}O$ , we are referring to benthic  $\delta^{18}O$ .

"As the reader, I'm left wondering why only oxygen isotopes were considered. Carbon isotopes are also available, why not include carbon isotopes in the analysis, as was done in the original Westerhold publication?"

We have clarified that our method can be applied to other proxies (Conclusion), including  $\delta^{13}$ C. While we focused on Westerhold et al. (2020)  $\delta^{18}$ O time series in this study, we have included an additional analysis focusing only on the Icehouse period and this demonstrates the broader applicability of our method in shorter samples.

"The authors discuss the varying resolution of the time series at length, which is helpful. However, I would be curious as to whether or not the resolution impacted the detection of break points. Is there any correlation between the detection of new breakpoints (discussed later in the manuscript) and the resolution of the time series? For example, just by visual comparison, it seems to me that the breakpoint observed in Coolhouse 1 in later sections might be related to a large change in resolution that occurs nearby. While this breakpoint isn't heavily interpreted here, this is an important point to understand if other researchers are to apply this method to their own data."

The resolution issue essentially reflects a bias-variance trade-off, where a higher binning frequency (i.e., higher resolution) reduces variance of the breakpoint estimates but may introduce some bias, whereas a lower binning frequency is more likely to increase variance but will be less prone to bias. This means that the primary effect of changing the resolution is on the constructed confidence intervals rather than on the location of the breakpoints themselves. This is evident in Figure 4 of the new manuscript, where a lower binning frequency generally leads to wider confidence intervals. Additionally, we observe strong stability in the estimated breakpoints across different binning frequencies, suggesting that resolution does not systematically impact breakpoint detection.

Regarding the estimated breakpoint in Coolhouse 1, it is indeed correct that this breakpoint is fairly close to a change in the resolution of the data. We agree that binned data reflect the quality of the original dataset, which can make it more challenging to accurately estimate breakpoints near or within data gaps. However, from a visual perspective, the breakpoint in Coolhouse 1 could also be influenced by the hump-shaped pattern that follows shortly after, between 17 and 14 million years ago.

"This sentence 'Furthermore, we recommend using binning frequencies 10 and 25 kyr as they result in the most consistent outcomes.' strikes me as rigid and somewhat unhelpful, as many records will not share the general time axis properties of CENOGRID. Is there a different way to describe your binning recommendation that's more flexible and/or applicable to other datasets?"

We acknowledge the referee's concern and have clarified that the choice of binning frequency should be tailored to the characteristics of the time series at hand. We have recommended selecting a frequency that maintains sufficient observations per bin while considering the dataset's length and resolution, and also quantified these for our application to make it more concrete. We have rephrased the sentence to make clear that our recommendation only refers to the CENOGRID benthic  $\delta^{18}$ O time series.

Certain technical choices need to be better explained given the audience of this journal. For example: 'To address these issues, we use the autocorrelation and heteroscedasticity consistent (HAC) covariance matrix estimator with prewhitening in our implementations.'. Perhaps this is standard fare in breakpoint analysis literature, but most paleoclimatologists won't be familiar with this procedure. Some explanation as to why this approach is suitable for this data is warranted here, as is mention of alternatives that were considered. In the same vein, it would be helpful to spend a little bit more time explaining information criteria. That is, expand upon "We use information criteria to estimate the number of breakpoints". What does this mean, why are they used, have they been used in paleoclimate contexts before, etc."

We appreciate the referee's feedback and have made these methodological choices more accessible to a paleoclimate audience. In the revised manuscript, we have expanded on the rationale for using the HAC covariance matrix estimator with prewhitening, explaining its role in addressing autocorrelation and heteroscedasticity, which are common in paleoclimate time series. We have also briefly discussed why our chosen approach is particularly suited for this application. The presence of autocorrelation and heteroscedasticity in paleoclimate time series has been considered in previous work focusing on ice core records, including Davidson et al. (2015) who considers heteroscedasticity, and Keyes et al. (2023) who investigate autocorrelation. Our manuscript focuses on paleoclimate records from ocean sediment cores, where similar challenges arise.

Furthermore, we have clarified the use of information criteria for selecting the number of breakpoint. While their use in paleoclimate research is still limited, studies like Valler et al. (2024) show they can be useful.

"Age uncertainty needs to be addressed somewhere in this paper. It doesn't need a full treatment, in that the method doesn't need to be modified to account for it, nor does it need to be included in the analysis, but discussion of how to include it in future studies is essential. Specifically I would be curious as to how this technique might be expanded to include the usage of age ensembles and how

choice of age modeling method might impact breakpoint detection. However, discussion of including age uncertainty directly would also be acceptable here."

We have included a brief discussion on age uncertainty and age ensembles (Section 3.3). While this is an important issue, we are not aware of any methodology in the paleoclimate literature that explicitly treats a timestamp as a random variable, with its variance representing the uncertainty of the timestamp. This challenge has also not been addressed within our framework, and incorporating the time-stamp uncertainty would require significant methodological developments. Such an extension involves advanced statistical techniques that are beyond the scope of this study. We have stated this clearly in the revised manuscript.

That being said, addressing age uncertainty is a crucial direction for future research. We have highlighted this in the revised manuscript and referenced relevant studies that discuss the issue (Telford et al., 2004; Franke and Donner, 2019) as well as a paper that explores aspects of transition detection in the presence of age uncertainty (Goswami et al., 2018). (Section 3.3)

The simulation study is a particular strength of this work. The authors thoroughly test their method across different data-generating processes, demonstrating its robustness to various forms of non-stationarity and serial correlation. This kind of rigorous testing is essential for establishing the reliability of statistical methods in paleoclimate contexts. I just wanted to make a note of that. We appreciate this positive feedback.

"When analyzing the possible presence of multiple breakpoints, I'm left desiring some kind of prescription as to how I should set the number of breakpoints. Certainly the claim that there are more than 5 statistically significant breakpoints in CENOGRID seems robust. However, the current analysis feels somewhat hand-wavy, with seven breakpoints being settled upon in a rather arbitrary way. In particular this statement needs to be expounded upon: 'The estimation results based on information criteria justify dividing the climate states Warmhouse II and Coolhouse II into two substates each at approximately 39.7 Ma and 10 Ma, respectively. This is supported by the presence of breakpoints estimated approximately at these timestamps in the estimations with seven or more breakpoints.'." We recognize the need for a clearer justification of the chosen number of breakpoints. As the results are somewhat ambiguous, we emphasize that the information criteria serve as a guidance tool rather than a strict statistical test for selecting the number of breakpoints. In the revised manuscript, we have provided a more detailed explanation of how information criteria inform breakpoint selection and have reframed the discussion to reflect their role in guiding, rather than testing, the choice of breakpoints (Section 3.2).

"The ending of this manuscript is far too abrupt. Potentially new breakpoints are discovered when

varying numbers of breakpoints are allowed, but what do they mean? A few climate events are referenced, but events themselves may or may not justify entirely new regimes. Much context is needed here, interpreting and explaining the presence of these novel breakpoints. While the authors are free to choose how to address this comment, I might suggest including a "Discussion" section, in which the primary results are emphasized, and an explanation/interpretation of these results is offered. Some of my other comments could probably be folded into this section as well."

We appreciate this suggestion and have included a new Discussion section (Section 4). Here, we discuss the paleoclimate implications of the estimated breakpoints and relate them to relevant literature. We have also softened the language around breakpoints to allow for the identification of both climate events and climate state transitions. While we do not claim to define entirely new climate regimes within the Cenozoic Era, our results statistically indicate that these time points mark shifts in the underlying dynamics of the time series, distinguishing them from other periods. Furthermore, we contextualize our findings within existing literature and provide insights for other researchers working with similar data, ensuring that our methodological contributions are framed within a broader paleoclimate context (Conclusion).

**Technical Comments:**

"The paper currently is a bit undercited. I suggest the authors go back through with a fine toothed comb and make sure they're citing existing literature wherever possible. In particular, all sections discussing  $\delta 18O$  interpretation should be thoroughly cited, particularly regarding ice volume effects, temperature relationships, etc. A couple of other key spots (non-exhaustive) that need citations include:

- 'The climatic transitions contain important information about variations in Earth's climate system' (here Tierney et al. 2020 is referenced, but some explanation of what is contained in that review along with additional citations is called for)
- 'Our approach contributes to the existing breakpoint detection methods in paleoclimate research" (cite breakpoint analysis in paleoclimate literature)'
- 'This breakpoint aligns with the Middle Eocene Climatic Optimum, a known climatic event' (cite original papers describing this event)
- 'Some of these breakpoints coincide with other climatic events, for instance, the Latest Danian Event at 62.2 Ma and the onset of the Miocene Climatic Optimum at 16.9 Ma' (cite original papers describing these events, not just Westerhold 2020)

We thank the referee for this detailed and helpful feedback. We have carefully reviewed the manuscript to ensure that all relevant literature is appropriately cited. In particular, we have strengthened the citations in Section 2.1 discussing  $\delta^{18}$ O interpretation, now referencing key studies on ice volume effects, temperature relationships, and other important factors.

Additionally, we have taken the following steps to address the referee's suggestions:

- We expanded the discussion of climatic transitions, providing a more detailed explanation of Tierney et al. (2020) and including additional citations highlighting the importance of Cenozoic climate states (Introduction).
- We contextualized our contribution to breakpoint detection in paleoclimate research by citing the reviews of Mudelsee et al. (2014) (benthic  $\delta^{18}$ O time series analysis) and Marwan et al. (2021) (nonlinear time series analysis), along with relevant applications (Introduction).
- We ensured that references to specific climatic events, such as the Middle Eocene Climatic Optimum, and the Miocene Climatic Optimum, are now supported by citations to original research studies rather than relying solely on secondary sources (Discussion). "

**3 Referee 2:**

"The authors apply a statistical approach to a reference paleoclimate dataset to evaluate shifts in climate states across the Cenozoic. They review existing methods and highlight the advantages of the breakpoint method and apply this approach on the Westerhold et al. 2020 dataset. They review the model and the impact of model parameterization on determined number of breakpoints. Ultimately, they established a similar number of breakpoints to Westerhold et al 2020 paper using this new approach and summarize their methodological approach and findings.

The application of new statistical approaches to assess time series and breakpoints is an important field of study but this contribution lacks significance beyond the application and usefulness of the method used. Although the paper is organized in structured manner, it is lacking background on other statistical assessments applied to assess Cenozoic or long-term changes in past climates and inclusion of a breath of paleoclimate specific references. Overall, this contribution mostly focuses on method development and application rather than implications of findings. Based on the above, it is difficult to see how it has led to a deeper understanding of Cenozoic climate.

I would suggest that this contribution needs to expands its discussion to outline how its findings enhanced our Cenozoic climate knowledge. Further there is a lack of background information needed to fully characterize the dataset evaluated, its attributes and limitations, and age model considerations. Below I have made some comments to develop the paper and give its finding wider significance.

**General comments:**

• Introduction lacks information on Earth's history and significance of these various climate states. A review of existing methods is useful but does not link to wider significance of work and

importance for paleoclimate field or mention how this analysis could provide new understanding or concepts for workers.

- Justification for use of Westerhold et al. (2020) dataset needs to be developed further. More information on benthic foraminifera used to construct record, limitations associated with age model differences, and potential interpretation of d180b record from past work. There is limited mention of d180b and its use to pinpoint climate states and whether these states align when using other similar datasets or depending on inclusion or exclusion of individual/regional d180b records does it impact the breakpoint determinations.
- The contribution needs to consider its finding and whether it has led to a deeper knowledge of Cenozoic Science. For instance, what framework can the climate transitions of the Cenozoic be categorized and how is this related to dynamics of the climate system? This will allow for the novelty of the method application to be developed in more detail. A thorough review of other approaches used to assess climate change in Cenozoic would be useful to include, going beyond this method, including tipping point analysis and frequency analysis to showcase how this co-eval alongside of the breakpoints."

We thank Referee 2 for the helpful and constructive comments. We appreciate the recognition of the importance of applying new statistical approaches to paleoclimate time series and breakpoint detection. As the referee notes, the primary focus of our contribution is methodological, but in the revised version, we have made a serious attempt to expand the paleoclimate context and implications of our results.

To accomplish this, we have added a co-author with expertise in Cenozoic climate history, which we believe has significantly improved the discussions of paleoclimate implications. While the methodological emphasis remains a core focus of the paper, we have made substantial efforts to strengthen the paleoclimate relevance and how our findings led to deeper understanding of the Cenozoic climate.

We have also expanded the motivation for selecting the CENOGRID dataset as our case study, explaining its strengths and suitability for illustrating our method (Introduction). Potential limitations, such as age model differences, regional variations, and other dataset characteristics, are also discussed briefly in Section 3.3, with reference to Westerhold et al. (2020) for more detailed discussions.

In response to the referee's specific comments, we have revised the manuscript in the following ways:

• We have expanded the Introduction to provide more background on Earth's climatic history,

the significance of Cenozoic climate states, and the need for statistical methods to characterize climate transitions.

- We have made a serious effort to integrate our findings within the paleoclimate science literature by incorporating key references:
  - Burke et al. (2018) who discuss how Cenozoic climate states serve as analogs for future warming scenarios.
  - Caballero and Huber (2013) who address the concept of state-dependent climate sensitivity.
  - Reviews by Mudelsee et al. (2014) and Marwan et al. (2021) which cover other statistical approaches for detecting climate transitions in paleoclimate time series.
- We have added a "Discussion" section (Section 4) where we interpret and contextualize our primary results, emphasizing their implications for understanding shifts in the dynamics of Earth's climate system during the Cenozoic. We also provide practical insights for researchers working with similar datasets and discuss the broader relevance of our findings (Conclusion).
- We have included a concise discussion of age model considerations and the challenges associated with age uncertainty, referencing key studies (Telford et al., 2004; Franke and Donner, 2019). An example of a transition detection method that considers age uncertainty is also referenced (Goswami et al., 2018). Although explicitly incorporating age uncertainty is beyond the current scope, we have highlighted this as an important direction for future research. (Section 3.3)
- To further demonstrate the flexibility and applicability of our method, we have added an analysis focusing solely on the Icehouse period (3.3 Ma to present) (Section 3.2).

**References**

Burke, K. D., Williams, J. W., Chandler, M. A., Haywood, A. M., Lunt, D. J., and Otto-Bliesner, B. L. (2018). Pliocene and eocene provide best analogs for near-future climates. *Proceedings of the National Academy of Sciences*, 115(52):13288–13293.

Caballero, R. and Huber, M. (2013). State-dependent climate sensitivity in past warm climates and its implications for future climate projections. *Proceedings of the National Academy of Sciences*, 110(35):14162–14167.

Davidson, J., Stephenson, D., and Turasie, A. (2015). Time series modeling of paleoclimate data. Environmetrics, 27:n/a-n/a.

- Franke, J. G. and Donner, R. V. (2019). Correlating paleoclimate time series: Sources of uncertainty and potential pitfalls. *Quaternary Science Reviews*, 212:69–79.
- Goswami, B., Boers, N., Rheinwalt, A., Marwan, N., Heitzig, J., Breitenbach, S., and Kurths, J. (2018). Abrupt transitions in time series with uncertainties. *Nature Communications*, 9(48).
- Keyes, N. D. B., Giorgini, L. T., and Wettlaufer, J. S. (2023). Stochastic paleoclimatology: Modeling the EPICA ice core climate records. *Chaos*, 33(9):093132. Special Collection: Theory-informed and Data-driven Approaches to Advance Climate Sciences.
- Marwan, N., Donges, J. F., Donner, R. V., and Eroglu, D. (2021). Nonlinear time series analysis of palaeoclimate proxy records. *Quaternary Science Reviews*, 274:107245.
- Mudelsee, M., Bickert, T., Lear, C. H., and Lohmann, G. (2014). Cenozoic climate changes: A review based on time series analysis of marine benthic  $\delta 18O$  records. Reviews of Geophysics, 52(3):333-374.
- Telford, R., Heegaard, E., and Birks, H. (2004). All age-depth models are wrong: but how badly? *Quaternary Science Reviews*, 23(1-2):1-5.
- Tierney, J. E., Poulsen, C. J., Montañez, I. P., Bhattacharya, T., Feng, R., Ford, H. L., Hönisch, B., Inglis, G. N., Petersen, S. V., Sagoo, N., Tabor, C. R., Thirumalai, K., Zhu, J., Burls, N. J., Foster, G. L., Goddéris, Y., Huber, B. T., Ivany, L. C., Turner, S. K., Lunt, D. J., McElwain, J. C., Mills, B. J. W., Otto-Bliesner, B. L., Ridgwell, A., and Zhang, Y. G. (2020). Past climates inform our future. Science, 370(6517):eaay3701.
- Valler, V., Franke, J., Brugnara, Y., Samakinwa, E., Hand, R., Lundstad, E., Burgdorf, A.-M., Lipfert, L., Friedman, A. R., and Brönnimann, S. (2024). Mode-ra: a global monthly paleoreanalysis of the modern era 1421 to 2008. Scientific Data, 11:36.
- Westerhold, T., Marwan, N., Drury, A. J., Liebrand, D., Agnini, C., Anagnostou, E., Barnet, J.
  S. K., Bohaty, S. M., Vleeschouwer, D. D., Florindo, F., Frederichs, T., Hodell, D. A., Holbourn,
  A. E., Kroon, D., Lauretano, V., Littler, K., Lourens, L. J., Lyle, M., Pälike, H., Röhl, U., Tian,
  J., Wilkens, R. H., Wilson, P. A., and Zachos, J. C. (2020). An astronomically dated record of
  Earth's climate and its predictability over the last 66 million years. Science, 369(6509):1383–1387.

---

## Author Response (AR2)

**Authors' response**

**August 2025**

**Editor:**

"Dear authors.

I am now in receipt of two reviews of your revised manuscript. Reviewer is satisfied with the work and supportive of immediate publication, while Reviewer 2 suggests additional work.

I must agree with Reviewer 2 that the implications of additional breakpoints need to be more carefully discussed. From a methodological standpoint, it comes as no surprise that estimating the number of breakpoints from information-theoretic criteria would lead to additional breakpoints compared to those identified previously in Westerhold et al (2020). This begs the question of the geological significance of these new breakpoints. As such, Reviewer #2's suggestion of incorporating different lines of evidence (in addition to benthic d180, 'd13Cb, relavant [sic] SST or BWT records with stages of cryosphere development + C3/C4' (the ratio of C3 to C4 plants)), is very well taken, and could shed light on whether transitions identified solely in d18O are geologically meaningful, or whether they are simply a methodological artifact. It may also be worth discussing how to integrate the Bai-Perron detection of breakpoints across multiple data streams (e.g. the ones cited above), though a formal investigation of this type is clearly out of scope for this paper.

Reviewer 2 also makes a number of useful, specific suggestions regarding additional references that would make the manuscript more current to the present state of the scholarship.

Finally, I agree that the choice of title could be improved, though for a different reason: while there is nothing wrong with using the mainstream word 'econometric', the issue I see is with the plural form 'methods'. This is misleading as only one such method (the Bai-Perron framework) is really used. Thus it would seem more truthful to choose the title 'Estimating breakpoints in the Cenozoic Era with the Bai-Perron framework', or 'Estimating breakpoints in the Cenozoic Era — an econometric approach', to avoid giving the impression that a whole array of econometric methods have been applied to the question.

In revising your manuscript, please make sure to observe the revisions checklist provided by the editorial office.

Best of lucks for your final revisions,"

We thank the editor for the constructive summary and for the opportunity to revise our manuscript further. We are pleased that Reviewer 1 supports immediate publication and that Reviewer 2 acknowledges substantial improvements in the revised version. We also appreciate the thoughtful comments from both the editor and Reviewer 2 on how to further clarify the implications of our findings and improve the broader paleoclimate context.

In line with the editor's suggestion, we have revised the title to: "Estimating breakpoints in the Cenozoic Era: An econometric approach." We agree that this better reflects the scope of the paper and avoids suggesting that multiple econometric methods were applied.

We also agree that a full formal analysis across multiple data streams is beyond the scope of this study. However, as suggested by Reviewer 2, we have now incorporated several relevant paleoclimate records, such as a  $\delta^{13}$ C stack, atmospheric CO2 concentration reconstructions, and sea-level reconstructions (Westerhold et al., 2020; Hönisch et al., 2023; Miller et al., 2020), into a new summary figure in the Discussion section. This provides geological context for the additional breakpoints and allows readers to better assess their significance. In addition, we have expanded the Conclusion to discuss how the Bai-Perron framework could be extended to analyze other proxy records in future work.

We have also addressed Reviewer 2's specific suggestions regarding additional references. The manuscript has been updated throughout to incorporate relevant recent work on recurrence analysis, Cenozoic climate records, and time series methods.

Finally, we have revised the reference list and appendices in accordance with the editorial office's checklist and instructions.

We hope that these revisions meet the expectations of both the editor and reviewers, and we thank you again for the valuable feedback.

**Referee 2:**

"Bennedsen et al. resubmission tackles applying econometric methods to estimate breakpoints. The previous version lacked context for wider paleoclimate science and was hard to discern the impact/importance of the new method for undersanding Cenozoic climate better. They have modified the manuscript across all sections to deepen the context and implications of the work.

Specifically the authors have improved the manuscript however more details are needed to justify some changes and include additional references/previous work to make this contribution exhaustive. Some aspects are related to additional references related to recurrence analysis, Cenozoic d18Ob records, use of econometric methods in time series analysis, and impacts of age uncertainty on findings.

Although the authors now have added a discussion to provide wider paleoclimate context which expands the application more, there is additional work needed to refine this section including restructuring and a summary figure.

With these revisions completed, this would make a good contribution for this journal.

Major comments:

Discussion section hints at potential additional break points however this is alongside of methods aspects. I would recommend adding a section in the results around 'additional breakpoints' to highlight method/criteria used, number of additional breakpoints identified, and then leave the implications of this for the discussion. Info about binning and regime length in discussion make it difficult to digest. I also recommend making a new figure to go along with this section.

Authors highlight there are potential additional breakpoints detectable in the time series. This section refers to Figure 5 only but highlights changes in carbon cycle, cooling and wider climate systematics. It would be useful to make an additional figure which includes d18Ob, d13Cb, relavant SST or BWT records with stages of cryosphere development + C3/C4 to showcase timing of possible new breakpoints and wider features. This could have the initial westerhold points and the new potential ones."

We thank the reviewer for their careful reading of the revised manuscript and for recognizing the improvements made across all sections. We appreciate the constructive feedback on how to further enhance the contextualization, structure, and clarity of the manuscript, and we agree that these revisions strengthen the overall contribution.

We have made the following revisions in response to the major comments:

- We have added further detail to support the interpretation of additional breakpoints, drawing on multiple lines of evidence from existing studies, including sea-level and  $CO_2$  reconstructions,  $\delta^{13}C$  data, and  $C_4$  vegetation shifts. While a full multi-proxy analysis is beyond the scope of this study, we now highlight how the Bai-Perron framework could be applied to other records in future work, and we have included new figures to support this discussion.
- We have restructured the Results and Discussion sections to improve clarity. Specifically, we have moved methodological and technical details (e.g., criteria used to identify breakpoints and binning procedures) to the Results section, leaving the Discussion to focus on interpretation and broader implications.

**Specific comments:**

• "Title has changed to include 'econometric' methods - however this term is not known in paleoclimate field and only mentioned at the end of the introduction very briefly defining it (line 83). I would recommend reconsidering title change or expanding on the use of these types of methods more widely or explaining why applying this from another field is worthwhile or novel or beneficial."

- We have revised the title to reflect that we are referring to a specific econometric method rather than the field more broadly. In addition, we have clarified the introduction of the econometric method earlier in the text and improved the explanation of its relevance to paleoclimate research.
- "Lines 30-33 it is worth mentioning that there are not only shifts in data set but the resolution is varying as well and reference accordingly"
  - We have chosen not to modify this passage. Variation in resolution is discussed at multiple points throughout the manuscript, and adding this detail here would, in our view, detract from the clarity of the message in the introduction.
- "Line 33- there are more recent publications to include as well (Mudelsee et al 2014) in addition to Zachos"
  - We have added a citation to Mudelsee et al. (2014) as suggested.
- "Line 45 are there other applications in the field of climate science that can be summarized used using recurrence analysis (e.g Liang et al 2025). It would be useful to provide exhaustive background on application of recurrence analysis and metrics used in this analysis in addition to westerhold"
  - We have included references to Liang et al. (2025) and Fischer et al. (2024) to reflect more recent applications of recurrence analysis in climate science. As we already provide a detailed description of recurrence analysis and its metrics on lines 43–80, and since our contribution lies in presenting an alternative approach rather than extending recurrence analysis itself, we prefer not to expand further on this background to maintain the focus of the paper.
- "Line 81 sets out the study but does not mention Westerhold dataset, could be useful to say this work applies approach on climate stack"
  - We thank the reviewer for noticing this. We have clarified this by explicitly stating that the proposed approach is applied to the Westerhold et al. (2020)  $\delta^{18}$ O stack.
- "Line 89 here the econometric framework is then then referred to as the Bai-Perron framework and using this term for the remaining. It is useful to introduce both econometric and Bai-Perron framework?"
  - We have restructured the end of the introduction to clarify the novelty of applying this framework to paleoclimate data. We chose not to introduce both terms separately, as the

Bai-Perron framework is thoroughly presented in the methodology section, and 'econometric' refers to a broader research field that is not easily defined in a few sentences.

- "Line 101- 'weight difference' consider defining as ratio of heavy to light for instance"
  - We have changed the definition accordingly.
- "Line 110-can you provide example resolution across key time intervals, the average resolution is only so useful as there are few records in early Cenezoic with orbital resolution"
  - We have added a sentence with examples of resolutions across the climate states defined by Westerhold et al. (2020).
- "Line 113- age model uncertainty is mentioned but also said it is not accounted for, can you add the biggest drivers of uncertainty here? It is later mentoned but would be worth to introduce reader to the issues here broadly also it is useful to highlight why it does not need to be addressed in this study. See lines 327-330."
  - Thank you for the suggestion. We have revised this section to include the primary sources of age model uncertainty, such as orbital tuning and sedimentation rates, and now clarify that the magnitude of this uncertainty is small relative to the duration of the climate states we estimate. We believe this provides sufficient context for why age model uncertainty is not explicitly addressed in our analysis.
- "Line 336-337 it is stated that the authors 'expect our main findings to be robust' despite age uncertainties across the Westerhold. Are the authors able to conduct a sensitivity test in some way to showcase this?"
  - We appreciate the reviewer's suggestion. While a formal sensitivity analysis would be valuable, it is beyond the scope of the present study.
- "Line 355- the authors mention the post-MECO cooling and refer to Bohaty & Zachos, 2003, however there are additional insights into the MECO and carbon cycle and temperature trends since this publication."
  - We have added more details to this part of the discussion by considering the findings of Henehan et al. (2020).
- "Line 368-denote average resolution for the record"
  - We have reported the average resolution of the record here.

- "Figure 1 and 3-add the geological time scale for reference as well as the climate states for reader context. This would help also understand changes in data density/resolution with time."
  - We have added geological time scales to the two figures.
- "Figure 3-caption should include more description of the data for instance including time interval, data resolution and number of core sites."
  - We have added more description of the data in the caption of Figure 3.
- "Heading 3.1-consider changing to 'breakpoint number sensitivity tests 'or more general than current"
  - We have changed the titles of Sections 3.1 and 3.2 to make them more accurate.

**References**

- Fischer, M. L., Munz, P. M., Asrat, A., Foerster, V., Kaboth-Bahr, S., Marwan, N., Schaebitz, F., Schwanghart, W., and Trauth, M. H. (2024). Spatio-temporal variations of climate along possible african-arabian routes of h. sapiens expansion. *Quaternary Science Advances*, 14:100174.
- Henehan, M. J., Edgar, K. M., Foster, G. L., Penman, D. E., Hull, P. M., Greenop, R., Anagnostou, E., and Pearson, P. N. (2020). Revisiting the middle eocene climatic optimum "carbon cycle conundrum" with new estimates of atmospheric pco2 from boron isotopes. *Paleoceanography and Paleoclimatology*, 35(6):e2019PA003713. e2019PA003713 2019PA003713.
- Hönisch, B., Royer, D. L., Breecker, D. O., Polissar, P. J., Bowen, G. J., and Henehan, M. J., et al. (2023). Toward a cenozoic history of atmospheric co2. *Science*, 382(6675):eadi5177. The Cenozoic CO2 Proxy Integration Project (CenCO2PIP) Consortium.
- Liang, J., Wang, Y., Zhang, S., Huang, C., Xu, E., and Zhang, Z. (2025). Astronomical Forcing of late oligocene to early Miocene Paleoclimate: A case study from the Northern South China Sea. Palaeogeography, Palaeoclimatology, Palaeoecology, 673:113007.
- Miller, K. G., Browning, J. V., Schmelz, W. J., Kopp, R. E., Mountain, G. S., and Wright, J. D. (2020). Cenozoic sea-level and cryospheric evolution from deep-sea geochemical and continental margin records. *Science Advances*, 6(20):eaaz1346.
- Mudelsee, M., Bickert, T., Lear, C. H., and Lohmann, G. (2014). Cenozoic climate changes: A review based on time series analysis of marine benthic  $\delta 18O$  records. Reviews of Geophysics, 52(3):333-374.

Westerhold, T., Marwan, N., Drury, A. J., Liebrand, D., Agnini, C., Anagnostou, E., Barnet, J. S. K., Bohaty, S. M., Vleeschouwer, D. D., Florindo, F., Frederichs, T., Hodell, D. A., Holbourn, A. E., Kroon, D., Lauretano, V., Littler, K., Lourens, L. J., Lyle, M., Pälike, H., Röhl, U., Tian, J., Wilkens, R. H., Wilson, P. A., and Zachos, J. C. (2020). An astronomically dated record of Earth's climate and its predictability over the last 66 million years. *Science*, 369(6509):1383–1387.

---

## Author Response (AR3)

**Authors' response**

**September 2025**

**Editor:**

"Dear authors,

After reading the manuscript, I am satisfied that the revisions have addressed the concerns and suggestions of the reviewers. I only recommend a few minor changes prior to publication:

We thank the editor for careful reading of our revised manuscript and for confirming that the review

We thank the editor for careful reading of our revised manuscript and for confirming that the reviewers' concerns have been addressed. We have implemented the suggested minor changes as follows:

L197+: 'aucorrelation occurs when current values correlate with past values, which is common in paleoclimate data due to long-term persistence in climate dynamics.' Here it should be added that taphonomic processes such as bioturbation also add to this aucorrelation in the proxy record.

We have added this clarification.

L304+: 'a few paleoclimate studies use information criteria for model selection, for example Valler et al 2024 show we can be beneficial'. This statement seems gratuitous as Valler et al 2024 were far from the first to use an IC in paleoclimatology. I recommend removing it altogether.

We agree with the editor and have removed this sentence.

L318-319: 'six and 12' and 'seven and 14'. Please use either numeric or textual representations, but do not mix and match.

We have chosen to use numeric representations and have made them consistent throughout.

L374-376: This argument is circular; you cannot use the Bai-Perron framework to validate the recurrence network analysis of Westerhold et al (2020), and use Westerhold et al (2020) to validate the Bai-Perron framework. Given that this framework is actually rooted in statistical theory and econometric practice, it seems like it would help buttress the results of Westerhold et al (2020). The intersection with the tipping points identified by Rousseau et al (2023) is also relevant here.

We have revised this part of the discussion accordingly to avoid circular reasoning and to highlight the relevance of Rousseau et al. (2023). L387: 'Five of these breakpoints closely match the five major transitions identified by Westerhold et al (2020).' Please clearly mark those transitions on Fig 7.

The transitions identified by Westerhold et al. (2020) were marked in the plot with dashed lines in the original manuscript. We have made these lines thicker and hopefully they are now presented more clearly in Fig 7.

L430: 'which Clark et al (2006) describe as a gradual transition occurring between 1.25 and .7 Ma'  $\rightarrow$  See also James et al (2024, 10.1029/2023PA004700) for a dynamical argument about a gradual transition."

We have added this reference.

**References**

Rousseau, D.-D., Bagniewski, W., and Lucarini, V. (2023). A Punctuated Equilibrium Analysis of the Climate Evolution of Cenozoic Exhibits a Hierarchy of Abrupt Transitions. *Scientific Reports*, 13:11290.

Westerhold, T., Marwan, N., Drury, A. J., Liebrand, D., Agnini, C., Anagnostou, E., Barnet, J.
S. K., Bohaty, S. M., Vleeschouwer, D. D., Florindo, F., Frederichs, T., Hodell, D. A., Holbourn,
A. E., Kroon, D., Lauretano, V., Littler, K., Lourens, L. J., Lyle, M., Pälike, H., Röhl, U., Tian,
J., Wilkens, R. H., Wilson, P. A., and Zachos, J. C. (2020). An astronomically dated record of
Earth's climate and its predictability over the last 66 million years. Science, 369(6509):1383–1387.